# EMERGENT COMMUNICATION AT SCALE

**Rahma Chaabouni**\*, **Florian Strub**\*, **Florent Altché** , **Corentin Tallec** , **Eugene Trassov** , **Elnaz Davoodi** , **Kory Mathewson** , **Olivier Tieleman** , **Angeliki Lazaridou** , **Bilal Piot**

DeepMind

## ABSTRACT

Emergent communication aims for a better understanding of human language evolution and building more efficient representations. We posit that reaching these goals will require scaling up, in contrast to a significant amount of literature that focuses on setting up small-scale problems to tease out desired properties of the emergent languages. We focus on three independent aspects to scale up, namely the dataset, task complexity, and population size. We provide a first set of results for large populations solving complex tasks on realistic large-scale datasets, as well as an easy-to-use codebase to enable further experimentation[1]. In more complex tasks and datasets, we find that RL training can become unstable, but responds well to established stabilization techniques. We also identify the need for a different metric than topographic similarity, which does not correlate with the generalization performances when working with natural images. In this context, we probe ease-of-learnability and transfer methods to assess emergent languages. Finally, we observe that larger populations do not induce robust emergent protocols with high generalization performance, leading us to explore different ways to leverage population, through voting and imitation learning.

## 1 INTRODUCTION

Language emergence is at the intersection of cognitive science and machine learning. From a cognitive science view, researchers have been looking at artificial agents as another expressive species to shed light on the source of linguistic regularities (Wagner et al., 2003; Guo et al., 2019; Chaabouni et al., 2021). From a machine learning view, language evolution is deemed as a promising direction to shape agents' representation and design interactive AI (Steels, 1997; Lazaridou et al., 2020).

Most of the literature in this field relies on different variants of the Lewis game (Lewis, 1969). There, a speaker network must describe an object to a listener network, which must then retrieve it among a set of other objects. To solve the game, the two agents need to settle on a communication protocol. While deep agents manage to solve the Lewis game, their communication protocols are usually degenerate, lacking core properties of human languages (Bouchacourt & Baroni, 2018; Chaabouni et al., 2019). In response to these findings, several works proposed ad-hoc solutions by constraining the game and agents' capacity (Kottur et al., 2017; Resnick et al., 2020). While reducing problem complexity is tempting, it can lead to unexpected outcomes and may miss general language emergence behaviors (Hayes, 1985). We take a different route and advocate that scaling up communication games is a prerequisite to building interactive AI (Baroni et al., 2017; Sutton, 2019) or modeling language evolution (Barsalou, 2008). Indeed, contrary to other machine learning communities, the emergent communication field mostly relies on small-scale games where only one speaker and one listener communicate about disentangled stimuli, which can hinder the generality of its conclusions. In this paper, we focus on three scaling dimensions: the dataset, the task complexity, and the population size. That is, we argue that making populations of deep agents communicate about larger and more realistic datasets and solve more complex tasks, is necessary if in the end we want these agents to interact with us or if we want to model human communication.

We study three properties of emergent languages: generalization, robustness to input noise, and ease of learning over transfer tasks, analyzing different facets of communication protocols. The proposed

---

\* Contributed equally.     Corresponding authors: {`rahmac`,`fstrub`,`piot`}@deepmind.com

[1] Source code: `github.com/deepmind/emergent_communication_at_scale`

scaling up remains computationally tractable, as most of the experiments can be done within a day on a single GPU. The source code is based on the Jaxline pipeline (Babuschkin et al., 2020).

Overall, our experiments provide a large spectrum of observations, both positive and negative. First, we found that scaling up the Lewis game quickly entails unstable RL optimization. We propose KL regularization (Geist et al., 2019) as an effective way to overcome this issue. Second, we observe that complexifying the task has two positive aspects: it better discriminates between methods and improves the generalization of the learned communication protocol. Third, we note no correlation between generalization and the widely used topographic similarity metric, which suggests that the latter is not adequate to assess the compositionality of the languages in our more complex setup. Instead, we take inspiration from the self-supervised learning literature and explore transfer learning as a new evaluation metric. Fourth, unlike what was observed in human communication (Gary Lupyan, 2010; Raviv et al., 2019a;b), we find little to no systematic benefit on emergent languages' properties when increasing the population size. Instead, we propose alternative methods to leverage populations, namely voting and imitation among speakers (Hester et al., 2018; Vecerik et al., 2017). In particular, we show that such population dynamics lead to more robust, productive, and in some cases easy-to-learn languages even when compared to our best seed without population, opening up new research opportunities. In the end, we expect that these observations, baselines, and good practices would allow the language emergence community to benefit further from deep RL advances and move the field closer to its goals of improving representations and interactive systems.

## 2 SETUP

### 2.1 DISCRIMINATION GAME

The discrimination game involves two players, *Speaker* and *Listener*, and is decomposed into three sequential stages. First, Speaker receives a target image $x$ and outputs a message $m$ that is sent to Listener. Second, Listener receives $m$ and selects an image $\hat{x}$ among a set $\mathcal{C}$ of different images containing the target image $x$. The set $\mathcal{C}$ is called *candidates*. Finally, the target image $x$ is revealed to Listener. If Listener selects the target image, then both players receive a positive reward of $1$, and $0$ otherwise. Speaker and Listener are parameterized by a set of parameters $\theta$ and $\phi$ respectively. The message $m(x, \theta) = (w_t(x, \theta))_{t=0}^{T-1}$ is a sequence of length $T$ of words $w_t(x, \theta)$. When no confusion is possible, we omit the dependence on $x$ and $\theta$ to simplify notations. A word $w_t$ is an element of a finite vocabulary set $\mathcal{W}$. For the image selected by Listener $\hat{x}(\phi, m, \mathcal{C})$, we also omit the dependence on $\phi$, $m$ and $\mathcal{C}$. Finally, the reward for Listener and Speaker is denoted by $R(x, \hat{x}) = 1_{x=\hat{x}}$.

**Communication in a population of agents.** A straightforward extension of the standard discrimination game is to train a population of agents. In this case, given a population of $N$ Speakers and $N$ Listeners, we sample $S$ Speakers and $S$ Listeners with replacement at each training step to construct $S$ random pairs; where each Speaker is paired with only one Listener.[2] All pairs observe the same examples and are trained independently to play the discrimination game as described in the 1-pair setting above. In our case, $S = N$ so that each agent is trained on average once per step, which allows a fair comparison with the baseline of 1-pair. At inference time, we take the first $P$ Speakers and Listeners and construct all the possible $P^2$ pairs to compute the accuracy, by averaging the rewards of all pairs as in (Mordatch & Abbeel, 2018).

**Exploiting the population.** The training dynamic of the population of agents described above does not take advantage of the diversity of the population. Indeed, we treat all agents in the same way. Yet, one motivation for the effectiveness of the population is the variety between Speakers to invent new structures and between Listeners to avoid over-specialization (Bromham et al., 2015). Prior works have looked at the impact of training agents with different dynamics (e.g., Ren et al., 2019; Guo et al., 2019; Cogswell et al., 2019; Li & Bowling, 2019). In this work, we introduce two different ways to exploit the population; on Speakers' side, we add imitation, and on Listeners' side, we allow Listeners to vote to get the final prediction. Mathematical details are found in Sec. 2.3.

*Imitation* training consists of two different steps. First, as described above, Speakers and Listeners are paired randomly for $M$ interaction steps to learn to communicate. Second, Speakers are trained for 1 imitation step as follows: we select the best Speaker (called "teacher") among $K$ randomly sampled Speakers without replacement, and use the remaining $K$-1 samples as "students". We then train all students in a supervised way to imitate the teacher. We alternate between the interaction

---

[2]In this work, we only consider populations with the same number of Speakers and Listeners.

and imitation steps. Both $M$ and $K$ are hyper-parameters (see Appendix A.5). Finally, *Voting* is only used at inference time, irrespective of the training mode. Here, instead of averaging $P^2$ pairs' rewards to compute accuracy, we now allow $P$ Listeners to vote to get a unique prediction. This is inspired by ensemble methods to reduce prediction errors (Polikar, 2012). More details on imitation and voting are reported in Sec. 2.3. For completeness, we also reproduce ease-of-teaching protocol as another effective way to exploit population in Appendix B.4 (Li & Bowling, 2019).

## 2.2 DATASETS AND NEURAL ARCHITECTURES

**Datasets.** We use the ImageNet (Deng et al., 2009; Russakovsky et al., 2015), and CelebA datasets (Liu et al., 2015), which respectively contain 1400k and 200k labelled images. CelebA contains 40 binary attributes per image, such as the presence of glasses, blond_hair. As the official CelebA splits have non-overlapping identities, it is impossible to perform regular identity classification on the test/valid sets. We thus construct a new split, where we shuffle images to have overlapping identities across splits. In both datasets, each image is center-cropped before being processed by a ResNet-50 encoder $f$ pretrained on ImageNet with the self-supervised method BYOL (Grill et al., 2020) to extract a representation of size 2048. Full implementation details are in Appendix A.6.

**Speaker architecture.** Speaker is a neural network that receives an image $x$ and outputs a message $m = (w_t)_{t=0}^{T-1}$. It is composed of a fixed (non-trainable) image encoder $f$ that transforms $x$ into an embedding $f(x)$, before being projected by a state adapter $c_\theta$ to an initial state of an LSTM (Hochreiter & Schmidhuber, 1997), $z_{-1,\theta} = c_\theta(f(x))$. The LSTM receives as input a word embedding $e_{t,\theta} = g_\theta(w_{t-1})$ and outputs the next state $z_{t,\theta} = h_\theta(z_{t-1,\theta}, e_{t,\theta})$. The first word embedding $e_{0,\theta} = g_\theta(\texttt{sos})$ is initialized with a start of sequence. The state $z_{t,\theta}$ is then fed to two different heads; the value head $v_\theta$ estimates the value of the expected reward knowing $z_{t,\theta}$ and the policy head $\pi_\theta$ computes the probability of the chosen word given $z_{t,\theta}$. Then, a sampling function $s$ picks the word $w_t = s(\pi_\theta(.|z_{t,\theta}))$. Finally $w_t$ is fed back to $g_\theta$ to produce the next word and so on until the maximum length $T$ is reached. At training, the word $w_t$ is sampled according to the policy whereas it is greedily selected at evaluation. Contrary to prior works (e.g., Kottur et al., 2017; Resnick et al., 2020), we do not restrict the channel capacity of agents to spur the emergence of systematic languages. Instead, we endow Speaker with a message space of size $|W|^T = 20^{10}$ ($|W| = 20$, $T = 10$), significantly larger than the number of available training examples.

**Listener architecture.** Listener is a neural network that receives Speaker's message $m$ and a set of image candidates $\mathcal{C}$, containing the target image $x$. It outputs the probability over each image $\tilde{x} \in \mathcal{C}$ of being the target image. Listener is composed of an LSTM cell $h_\phi$, which is initialized to the null vector $z_{-1,\phi}$. The message $m$ is decoded by processing the sequence of word embeddings $e_{t,\phi} = g_\phi(w_t)$ through the LSTM such as $z_{t,\phi} = h_\phi(z_{t-1,\phi}, e_{t,\phi})$. The state $z_{T-1,\phi}$ is then fed to the network $p_\phi$ to output $p_{m,\phi} = p_\phi(z_{T-1,\phi})$. In parallel, each image $\tilde{x}$ is projected through the encoder $f$ and the network $t_\phi$ to obtain the image embedding $t_{\tilde{x}} = t_\phi(f(\tilde{x}))$. Both message and image embeddings are then compared through a score function $\texttt{score}(m, \tilde{x}, \phi) = \cos(\frac{p_{m,\phi}}{\|p_{m,\phi}\|_2}, \frac{t_{\tilde{x}}}{\|t_{\tilde{x}}\|_2})$. The scores over all images are normalized via a softmax to get a probability $\pi_\phi(.|m, \mathcal{C})$. Finally, Listener selects an image by taking the best guess according to $\pi_\phi$, i.e. $\hat{x} \in \arg\max_{\tilde{x} \in \mathcal{C}} \pi_\phi(\tilde{x}|m, \mathcal{C})$.

Architecture details, hyper-parameters and graphical descriptions are provided in Appendices A.2 and A.4. All experiments are run over 10 seeds.

## 2.3 OPTIMIZATION

**Speaker training and loss.** The goal of Speaker is to optimize the message $m$ sent to Listener such that the expected reward of the game is the highest possible. This can be framed as a sequential decision making problem where the decision is the choice of each word $w_t$. Therefore, following the policy gradient approach with a baseline (Sutton et al., 2000), we train Speaker's network by (i) minimizing a value loss $L_V(\theta)$ to make the value head $v_\theta(z_{t,\theta})$ fit the expected reward over a batch $\mathcal{X}$: $L_V(\theta) = \frac{1}{|\mathcal{X}|} \sum_{x \in \mathcal{X}} \sum_{t=0}^{T-1} (R(x, \hat{x}) - v_\theta(z_{t,\theta}))^2$, (ii) maximizing the expected reward through minimizing the policy loss $L_\pi(\theta) = \frac{1}{|\mathcal{X}|} \sum_{x \in \mathcal{X}} \sum_{t=0}^{T-1} \texttt{sg}(R(x, \hat{x}) - v_\theta(z_{t,\theta})) \log(\pi_\theta(w_t|z_{t,\theta}))$, where $\texttt{sg}(.)$ is the stop gradient function.

In addition, it is common practice in RL and emergent language literature to minimize an entropy loss $L_{\mathcal{H}}(\theta)$ encouraging to explore other choices of words by Speaker (Mnih et al., 2016; Williams & Peng, 1991; Espeholt et al., 2018). Finally, several deep RL (Schulman et al., 2015; 2017) and

theoretical RL papers (Geist et al., 2019; Vieillard et al., 2020a;b) argued that minimizing a KL loss $L_{\text{KL}}(\theta)$ between the online policy $\pi_\theta$ and a target policy $\pi_{\overline{\theta}}$ instead of or in addition to entropy regularization could be beneficial for better final performance as well as stabilizing the learning. The policy $\pi_{\overline{\theta}}$ is obtained by doing an exponential moving average of the weights $\theta$ over training: $\overline{\theta} \leftarrow (1-\eta)\theta + \eta\overline{\theta}$ where $\eta$ is the exponential moving average parameter. The KL minimization encourages the online policy to change slowly and smoothly.

To sum up, the speaker training loss $L(\theta)$ on a batch of images $\mathcal{X}$ is: $L(\theta) = L_V(\theta) + L_\pi(\theta) + \alpha L_{\mathcal{H}}(\theta) + \beta L_{\text{KL}}(\theta)$, where $\alpha$ and $\beta$ are hyper-parameters.

**Listener training and loss.** Listener is also trained to maximize the reward but acts by predicting the best guess for the game $\hat{x} \in \arg\max_{\tilde{x} \in \mathcal{C}} \pi_\phi(\tilde{x}|m, \mathcal{C})$. For each image $x$ in a batch $\mathcal{X}$, a set of image candidates $\mathcal{C}$ is sampled randomly (uniform without replacement over $\mathcal{X} \setminus \{x\}$) chosen in $\mathcal{X}$. When $|\mathcal{C}| = |\mathcal{X}|$, we take $\mathcal{C} = \mathcal{X}$. Finally, Listener's goal is to retrieve $x$ among $\mathcal{C}$, i.e. outputting a probability $\pi_\phi(\tilde{x}|m, \mathcal{C}) = 1$ if $x = \tilde{x}$ and $\pi_\phi(\tilde{x}|m, \mathcal{C}) = 0$ otherwise. Therefore, we use a multiclass classification loss where the correct class is the index of $x$ in the set of candidates $\mathcal{C}$ also called InfoNCE loss (van den Oord et al., 2018): $L(\phi) = -\frac{1}{|\mathcal{X}|} \sum_{x \in \mathcal{X}} \log\left(\pi_\phi(x|m, \mathcal{C})\right)$.

**Imitation training among a group of Speakers.** In a training imitation step, a group of $K$ speakers among the total population of $N$ speakers is sampled without replacement. Among those $K$ speakers, one speaker plays the role of the teacher and $K - 1$ play the role of the students. To choose the teacher, we compute, for each sampled speaker $i$, the exponential moving average of the accuracies over each batch on which the speaker $i$ has been trained on, $\sigma_i$. Then the teacher is simply the speaker with the highest $\sigma_i$. For convenience, we respectively note $\theta_T$ and $\theta_S$ the parameters of the teacher and student. A student $\theta_S$ is trained on a batch of data $\mathcal{X}$ by imitating the messages of the teacher $\theta_T$ with the following loss: $L_I(\theta_S) = -\frac{1}{|\mathcal{X}|} \sum_{x \in \mathcal{X}} \sum_{t=0}^{T-1} \log \pi_{\theta_S}(w_t(x, \theta_T)|x, z_{\theta_S, t})$, where $L_I(\theta_S)$ is a cross-entropy loss to encourage a student to output the same words as the teacher.

**Listener's voting at inference time.** At inference time, we can use all listeners $(\phi_i)_{i=0}^{N-1}$ of the population as an ensemble of networks. Together, they vote for a joint guess $\hat{x}$ over a set of candidates $\mathcal{C}$ of images for a message $m$ coming from a speaker. One simple way consists in averaging the score probabilities of the listeners and taking the best guess of this average. Formally, for each listener $\phi_i$, each message $m$ and batch $\mathcal{X}$, we have the score probability $\pi_{\phi_i}(.|m, \mathcal{C})$. Then the choice $\hat{x}(m, \mathcal{C}, (\phi_i)_{i=0}^{N-1})$ of the population of listeners for the message $m$ and the set $\mathcal{C}$ is the best guess of the average distribution which is $\hat{x}(m, \mathcal{C}, (\phi_i)_{i=0}^{N-1}) = \arg\max_{\tilde{x} \in \mathcal{C}} \frac{1}{N} \sum_{i=0}^{N-1} \pi_{\phi_i}(\tilde{x}|m, \mathcal{C})$.

More details about how we derive the different losses are found in Appendix A.3.

## 2.4 LANGUAGE PROPERTIES

**Generalization.** It measures the ability of agents to communicate about never-seen inputs. To compute generalization, we simply report test accuracy. In the simple case where inputs are constructed by disentangled attributes, test inputs are new combinations of previously seen attributes at training.

**Robustness.** We report agents' drop of accuracy when faced with noisy inputs relatively to clean inputs at test time. To construct noisy stimuli, we add a Gaussian noise for each batch, with mean 0 and a standard deviation that is half of the standard deviation of the batch. Note that we sample different noises for Speakers' and Listeners' inputs. That is, while Listeners are trained to find the exact input of Speakers at train time, they are now required to uncover a different input at eval time.

**Topographic Similarity (TopSim).** *TopSim* (Brighton & Kirby, 2006) is used as a proxy for compositionality (e.g., Li & Bowling, 2019; Lazaridou et al., 2018), and it is widely considered crucial for generalization (Fodor & Lepore, 2002; Marcus, 2003). *TopSim* tests whether close objects in the input space are described by close messages by computing the Spearman correlation between the pairwise distances in the input and message spaces. Following prior works, we use the edit-distance and the cosine-distance in the message and input spaces respectively.

**Ease and transfer learning (ETL).** It captures how fast and well the emergent language is transmitted to new Listeners performing a distinct task. It is similar to the ease-of-teaching (Li & Bowling, 2019) to the exception that Listeners are trained to solve a different task from the initial one that the emergent language was optimized for. This task transfer is inspired by the self-supervised linear evaluation protocol (Chen et al., 2020). To measure *ETL*, we take $\min(N, 5)$ fixed Speakers from the population after convergence, and feed their deterministic language to 3 newly initialized Lis-

teners. We argmax from Speakers' distribution to construct the deterministic languages. Finally, we show how fast and well new Listeners perform in a given task when trained on fixed languages by reporting the training curve for 10k steps. We look at the ease of learning when Listeners are trained to solve harder discrimination, classification, and image reconstruction tasks. Hence, *ETL* not only captures the generality of the language to new Listeners but also its ability to transfer to new tasks.

## 3 EXPERIMENTS

### 3.1 TASK SCALING UP: INCREASING THE NUMBER OF CANDIDATES

In the emergent communication literature, agents are typically required to discriminate between less than 20 objects (Mu & Goodman, 2021) or reconstruct hand-crafted attributes (Rita et al., 2020). Yet, this simple training setting was shown to sometimes lead to degenerate communication protocols (Bouchacourt & Baroni, 2018). As highlighted by (Dessì et al., 2021; Guo et al., 2021), the Lewis game is a discrete variant of Contrastive Predictive Coding in the self-supervised learning field (van den Oord et al., 2018). There, it was observed that the intermediate representation quality improved when increasing the number of candidates (He et al., 2019). We hence look at the impact of the number of candidates $|\mathcal{C}|$ on languages generalization and the technical challenges it introduces. In this subsection, we consider the 1-pair setting trained on ImageNet while varying $|\mathcal{C}|$.

**Scaling up task difficulty requires to carefully tune optimization.** We first train agents without KL regularization ($\beta = 0$) as it is commonly done in the field (see e.g., Strub et al., 2017; Cao et al., 2018; Lu et al., 2020, among many others). As displayed in Fig. 1(a), the training becomes unstable when increasing the number of candidates from 20 to 1024 candidates. The mean train accuracy drops from 98.9% to 76.0%, and we observe a large variance across seeds. This high variance demonstrates how crucial it is to run multiple seeds to state robust conclusions. A remedy against such instability in RL consists in adding a KL regularisation $L_{\mathrm{KL}}$ loss between the online policy and a target policy as described in Sec. 2.3. Such a solution has yet not been explored in the emergent communication field. We add a KL regularization with a coefficient $\beta = 0.5$ and display the results in Fig. 1(b). We note that, with this better regularization, 20 and 1024 candidates converge to 99.8% and 98.3% mean accuracy respectively with small variance during training. Adding $L_{\mathrm{KL}}$ thus stabilizes the training and leads to better performances. We illustrate further the effect of this regularization across multiple sweeps in Appendix B.3.

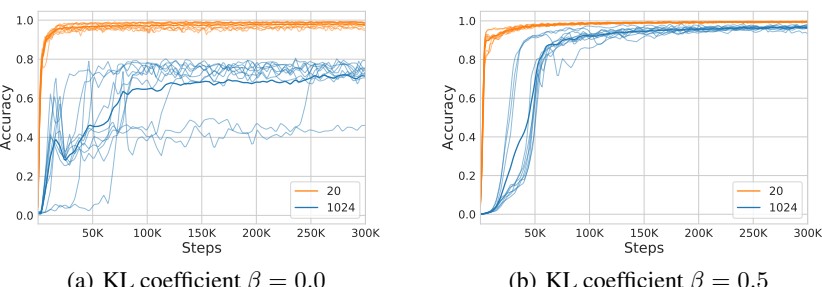

|     |     |
| :---: | :---: |
| (a) KL coefficient $\beta = 0.0$ | (b) KL coefficient $\beta = 0.5$ |

Figure 1: Training curves per task complexity (20 vs. 1024 candidates) w/ and w/o KL regularization on ImageNet. Thin lines are the accuracy of 10 seeds while thick lines represent the mean.

**Scaling up task difficulty is necessary to differentiate representations and enhance protocol's generalization.** In Fig. 2(a), we train 1-pair of agents on different number of candidates ranging from 64 to 1024 and evaluate them on 16 candidates.[3] Note that, in the literature, the number of distractors rarely exceeds a dozen both at train and eval times (Mu & Goodman, 2021; Li & Bowling, 2019). In such settings, our experiments do not provide any interesting conclusions: all methods are above 99.7% test accuracy and within standard deviation. However, when evaluating the same communication protocols now on harder tasks with 1024 candidates, as shown in Fig. 2(b), we note that these protocols are actually different. In particular, agents achieve an accuracy of 87.36%, 93.25%, 96.09% when trained on 64, 256, and 1024 candidates respectively, differentiating the various representation's generalization capacity. While expected in light of the recent self-supervised results (Chen et al., 2020) - harder training tasks lead to better representation and harder evaluation tasks better discriminate algorithms - these results clearly illustrate how current emergent language

---

[3]We kept the training batch size to 1024, and only varied the number of candidates in the InfoNCE loss.

experiments may be ill-scaled to have robust conclusions. We report further experiments in Appendix B.5. In the following, we use 1024 candidates at train and eval to fully leverage our findings.

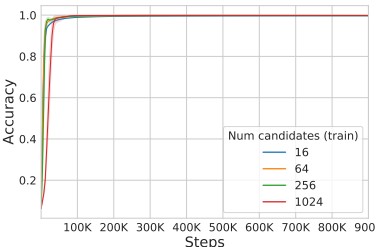 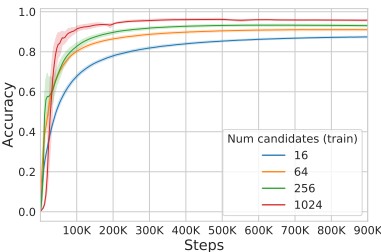

(a) Num candidates (eval): 16         (b) Num candidates at (eval): 1024

Figure 2: Test accuracy for different $|\mathcal{C}|$ at train (lines) and eval (subplot) times on ImageNet. (a) Easy eval task: no differentiation between methods. (b) Hard eval task: more candidates at training induces higher test accuracy. Shaded region represents the standard deviation across 10 seeds.

## 3.2 DATASET SCALING UP: RETHINKING EVALUATION

In most emergent communication works, deep agents are situated in a simple disentangled environment where objects are one-hot vectors (Kottur et al., 2017; Li & Bowling, 2019). As a result, the state space rarely goes beyond a few thousands of discrete samples. Besides, such representations are unambiguous as they are inherently compositional. We argue that such environments may be too simplistic to reach complex or diverse communication behaviors. In contrast, the major success stories in machine learning collectively prove the importance of training neural networks on a rich and large amount of data to emulate complex distributions (Brown et al., 2020; Krizhevsky et al., 2012; Sutton, 2019). Furthermore, if the goal is to reproduce human interaction, it is fundamental to incorporate complex stimuli to develop advanced concepts in language evolution (Miller & Johnson-Laird, 1976; Barsalou, 2008). In this subsection, we scale up the input space by using large natural image datasets, namely ImageNet and CelebA. In particular, we investigate further emergent language properties in the same setting of 1-pair of agents while varying $|\mathcal{C}|$ at train time.

| *TopSim* | **ImageNet** | **CelebA** | |
|---|---|---|---|
| Training | Image | Image | Attributes |
| $|\mathcal{C}| = 16$ | $15.50_{\pm 0.61}$ | $\mathbf{31.92_{\pm 1.32}}$ | $\mathbf{14.22_{\pm 1.36}}$ |
| $|\mathcal{C}| = 64$ | $\mathbf{19.01_{\pm 0.35}}$ | $28.7_{\pm 0.56}$ | $12.23_{\pm 0.76}$ |
| $|\mathcal{C}| = 256$ | $17.06_{\pm 1.09}$ | $30.69_{\pm 1.04}$ | $12.29_{\pm 1.27}$ |
| $|\mathcal{C}| = 1024$ | $16.52_{\pm 1.16}$ | $30.21_{\pm 1.22}$ | $13.49_{\pm 0.89}$ |

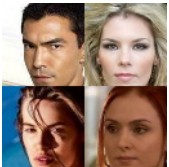 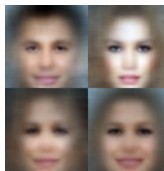

Figure 3: (left) *TopSim* (x100) with different representation of the input space (Image and Attributes) for different training difficulties, $|\mathcal{C}|$. We do not observe a correlation between *TopSim* and $|\mathcal{C}|$, and thus generalization. $\pm$ denotes 1 standard error of the mean. (right) Example of predicted images by new Listener trained on reconstruction task given a message of 1-pair for $|\mathcal{C}|$=1024. Reconstructions are expected to be blurry, as we are regressing the mean of all faces associated to one message.

***TopSim* may fall short with natural images.** We compute *TopSim* using the pretrained image logits on ImageNet and CelebA as input representation while varying the task complexity at train, $|\mathcal{C}|$. Results are provided in Fig. 3(left). We do not observe a consistent pattern between *TopSim* and $|\mathcal{C}|$. Furthermore, we note a non-significant (p-value>0.19) Spearman correlation of $-0.09$ and $0.08$ between *TopSim* and generalization for ImageNet and CelebA respectively. Although pretrained logits are excellent linear features (Grill et al., 2020), they may not be compositional. We thus reiterate the evaluation by using the attributes from CelebA as input representation to compute *TopSim*. In that case too we do not note any significant correlation (p-value=0.13). Similar findings have been observed in (Andreas, 2019; Chaabouni et al., 2020). This could be due to different reasons. First, agents could be communicating in a compositional way, to support good generalization, by encoding unexpected compositional features that are not labeled, like forehead shape or smile angle. That is, human-labeled features may differ from the agents' ones. Second, *TopSim* relies on strong assumptions, such as the chosen distance, and the use of a linear correlation. Such hypotheses may not hold when moving from artificial to realistic data. Finally, due to the large channel capacity, agents could be using synonyms to generalize with low *TopSim* values. In sum, our results demonstrate that

Table 1: Evaluation of Speakers on multiple settings on CelebA in %. $\pm$ denotes one standard error of the mean. We report final accuracy for all *ETL* tasks but Recons. where we report the final loss.

| *Metrics* | **Generalization** | **Ease and Transfer Learning** | | | |
|---|---|---|---|---|---|
| Training | Accuracy ↑ | Discr. ↑ | Identity ↑ | Attributes ↑ | Recons. ↓ |
| $|\mathcal{C}| = 16$ | $74.50_{\pm 0.96}$ | $56.31_{\pm 1.07}$ | $15.52_{\pm 0.90}$ | $86.99_{\pm 0.10}$ | $2448_{\pm 16}$ |
| $|\mathcal{C}| = 64$ | $80.19_{\pm 0.63}$ | $66.53_{\pm 1.09}$ | $23.12_{\pm 1.06}$ | $87.91_{\pm 0.11}$ | $2419_{\pm 12}$ |
| $|\mathcal{C}| = 256$ | $85.03_{\pm 0.72}$ | $76.61_{\pm 1.14}$ | $33.72_{\pm 1.52}$ | $88.81_{\pm 0.16}$ | $2355_{\pm 13}$ |
| $|\mathcal{C}| = 1024$ | $\mathbf{89.00_{\pm 0.48}}$ | $\mathbf{83.57_{\pm 0.78}}$ | $\mathbf{44.00_{\pm 1.26}}$ | $\mathbf{90.08_{\pm 0.13}}$ | $\mathbf{2351_{\pm 14}}$ |

*TopSim* is not a good predictor of communication protocol generalization for large-scale settings. Instead, we investigate the information content via *ETL*.

**ETL is a robust metric to evaluate languages with natural images.** *ETL*, contrary to *TopSim*, evaluates how useful the emergent language is for a target task beyond overfitting. As such it is an important metric, since the goal of emergent communication is to endow agents with communication skills, rather than solving a specific game. Results are reported in Table 1 over distinct tasks on CelebA. We observe that, even though *ETL* only looks at the information content and not at compositionality, it is a better predictor for language generalization performances than *TopSim*, with a strong and significant correlation >0.90 (p-value~0) for all considered tasks. As a visual sanity check, we generate predicted images of the reconstruction *ETL* task in Fig. 3(right). There, a new Listener is trained to minimize an $l^2$ loss with the target images given the precomputed messages. We observe that gender, makeup, and hair color are fairly encoded, while others such as the smile are not. More samples and training details are available in Appendix B.1. Interestingly, some *ETL* tasks, like discrimination or reconstruction, do not depend on (human-)predefined input representations, and are all more robust to channel capacity and linearity assumptions, as opposed to *TopSim*, which makes *ETL* a more general evaluation metric.

**Zero-shot dataset transfer highlights emergent protocols' differences and limits.** Scaling up to natural image opens a large diversity of visual distributions that we can leverage by using zero-shot dataset transfer (Yogatama et al., 2019; Lambert et al., 2020). We hence measure how generic and high-level the emergent protocols are. We take agents trained on ImageNet with $|\mathcal{C}|$=1024, and evaluate them on CelebA in a zero-shot fashion. There, the mean accuracy dropped from 95.96 to 36.73, and the standard deviation raised from 0.49 to 12.86. Noticeably, the most different agents obtain a zero-shot accuracy of 27.88 and 68.49, while their initial ImageNet accuracy only differed by 0.23. Firstly, such a standard deviation gap suggests that different emergent protocols have emerged between agent pairs. Secondly, the accuracy drop indicates that the emergent protocols remain specific to the initial data-distribution, and no systematic language has yet emerged.

### 3.3 POPULATION SCALING UP: EXPLOITING THE POPULATION DYNAMIC

The emergent communication literature often considers a single pair of agents (Lazaridou et al., 2017; Foerster et al., 2016). However, there is evidence in the multi-agent literature that such a setting may lead to extreme co-adaptation and overfitting (Lanctot et al., 2017). One counter-measure consists of sampling agents within a population (Jaderberg et al., 2018). From a linguistic perspective, different large-scale corpora analyses and human simulations support the importance of the population in shaping our languages (Gary Lupyan, 2010; Bromham et al., 2015; Raviv et al., 2019a;b). In this context, Raviv et al. (2019a) show that larger populations develop more systematic languages through human experimentation. Nonetheless, a few artificial simulations tentatively consider population sizes up to 32 agents, but with mixed results about their advantage (Mordatch & Abbeel, 2018; Tieleman et al., 2018; Graesser et al., 2019; Cogswell et al., 2019). In this part, we consider up to 100 agents. Specifically, we explore the impact of the population size on languages properties when we deal with more complex tasks ($|\mathcal{C}|$=1024 at train and eval) and realistic inputs (CelebA & ImageNet) while varying the population size $N \in \{1, 10, 50\}$ pairs.

**The best single-pair seed should be the baseline against population.** A few works in the emergent communication framework looked at the benefit of the population by comparing $2N$-agent's performances to the 2-agent baseline (Cogswell et al., 2019). However, the former introduces $\times N$ more parameters, which could be responsible for its slight observed benefit. Here, we consider instead the "best 1 pair" setting to investigate whether it is computationally advantageous to train a population of size $N$ rather than training $N$ independent pairs. That is, for a given computational budget, we ask whether it is beneficial to train agents within a population as opposed to $N$ pairs in parallel, as

Table 2: Different language properties on CelebA and ImageNet datasets, in %. For each setting we report the mean over 10 seeds. $\pm$ denotes 1 standard error of the mean.

| Setting | CelebA | | ImageNet | |
|---|---|---|---|---|
| | Generalization $\uparrow$ | Robustness $\downarrow$ | Generalization $\uparrow$ | Robustness $\downarrow$ |
| best 1 pair | 90.73 | 35.82 | **97.55** | 27.83 |
| 1 pair | $89.00_{\pm 0.48}$ | 37.90 | $96.09_{\pm 0.21}$ | 18.90 |
| 10 pairs | $91.06_{\pm 0.23}$ | 37.56 | $95.78_{\pm 0.29}$ | 14.58 |
| 50 pairs | $90.69_{\pm 0.61}$ | 38.87 | $95.29_{\pm 0.34}$ | 15.21 |
| 10 pairs + imitation | $91.84_{\pm 0.31}$ | 35.47 | $96.79_{\pm 0.12}$ | 13.46 |
| 10 pairs + imitation + vote | $92.19_{\pm 0.30}$ | 35.21 | $96.95_{\pm 0.11}$ | 13.27 |
| 50 pairs + imitation | $92.82_{\pm 0.51}$ | 32.69 | $96.70_{\pm 0.16}$ | 12.92 |
| 50 pairs + imitation + vote | $\mathbf{93.13}_{\pm \mathbf{0.50}}$ | **32.40** | $96.85_{\pm 0.15}$ | **12.75** |

also done in (Tieleman et al., 2018). "Best 1 pair" is constructed by selecting the best seed among the 10 seeds of "1 pair", based on validation accuracy, offering a fairer baseline *specifically* for the "10 pairs" setting. Results are shown in Table 2 and Fig. 4. We observe different behavior of "best 1 pair" across datasets. Particularly, if it always achieves better generalization and *ETL* discrimination compared to "1 pair", it shows, for ImageNet only, the worst *Robustness* and *ETL* classification. We hypothesize that in that specific case, the emergent protocol overspecializes to the task. Still, "best 1 pair" outperforms the standard baseline "1 pair" 6/8 times, which makes it a stronger baseline. In the following, we thus always compare the population setting with "best 1 pair" for a fair evaluation.

**Population size does not lead to a systematic advantage.** We look at the language generalization and robustness performances when increasing the population size. Results in Table 2 do not show a clear trend between population size and these metrics. For example, for both datasets, "50 pairs" achieves lower *Generalization* and *Robustness* than "10 pairs". Furthermore, "best 1 seed" always achieves better *Generalization* compared to the largest population setting "50 pairs". In Fig. 4, we look at *ETL* for a further investigation of the languages' properties. CelebA results exhibit a benefit of the population. However, this benefit does not always correlate with population size where "10 pairs" outperforms "50 pairs" on the discrimination task. This non-systematic benefit is also found when experimenting with ImageNet. Again, our results suggest no benefit of the population size in the current setting. We thus explore new approaches to leverage the population dynamics and improve the representation of the emergent protocol in the following.

**A better use of the population is needed to see larger benefit.** As population size does not improve the emergent language in itself, we took advantage of the richness of the population by considering imitation at training and vote at inference. In Table 2, we observe that both imitation and vote introduce a systematic improvement compared to the standard training in a population and lead to better performances compared to "1 pair" in all metrics. Moreover, when considering the stronger baseline "best 1 pair", the latter only outperforms the imitation and vote settings 1/4 times (*Generalization* for ImageNet). However, as noted previously, this case is an over-specialized protocol. Furthermore, if both imitation and vote introduce a systematic improvement of the protocol, their benefit is even more noticeable for distribution-shifted settings. For example, the imitation for 50 pairs adds a relative improvement of 12.8% in *Robusteness* compared to 2.3% for *Generalization* on CelebA. The same observation holds for the vote but with a lower degree. Note that if vote induces only a slight benefit, it does not incur any training cost. When considering *ETL* on discrimination in Figs. 4(a)&4(c), we observe that imitation leads to a systematic improvement. In CelebA, if the standard population was already beneficial, adding the imitation strengthens the results. In ImageNet, "10/50 pairs + imitation" perform similarly at the end of training than the overspecialized "best 1 pair" while having more stable optimization and it is considerably better than the standard training w/ and w/o population. Yet, the results are more nuanced when considering *ETL* over classification in Fig. 4(b)&4(d). Specifically, while CelebA experiments suggest an advantage when training a population of agents w/o imitation as opposed w/ imitation or "1 pair", this pattern is absent with ImageNet. There all settings with population are on par. In sum, both considered dynamics outperform in most cases the strong "best 1 pair" baseline (7/8 times), highlighting their viability.

# 4 DISCUSSION AND CONCLUSION

The emergent communication framework has been extensively studied for decades before the successes of deep RL. In this context, many works have revisited this framework through deep RL settings (e.g., Kirby 2002 *vs.* Ren et al. 2019, Myers-Scotton et al. 2002 *vs.* Graesser et al. 2019, or

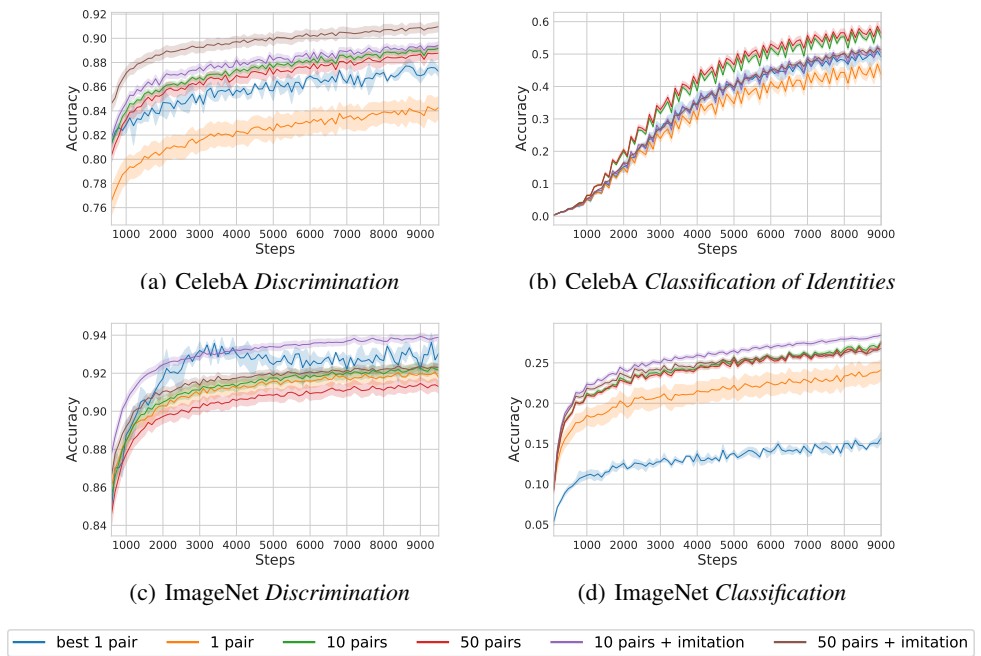

(a) CelebA *Discrimination*

(b) CelebA *Classification of Identities*

(c) ImageNet *Discrimination*

(d) ImageNet *Classification*

best 1 pair — 1 pair — 10 pairs — 50 pairs — 10 pairs + imitation — 50 pairs + imitation

Figure 4: *ETL* per datasets and tasks. The results are averaged across $\min(N, 5)$ Speakers, 3 new-born Listeners' over 10 different seeds. The shaded region represents the standard deviation.

Batali 1998 *vs.* Choi et al. 2018). However, the experimental settings have barely evolved for twenty years. For instance, Kirby (2002) and Ren et al. (2019) both used a binary input vector of size eight, and only a few papers try to go beyond such artificial input spaces or simple tasks (Lazaridou et al., 2017; Havrylov & Titov, 2017; Dessì et al., 2021). While computational constraints may have been a limiting factor, we show that our setting is possible with widely available hardware in Appendix B.2. For example, Table 1 requires approximately 400 hours of compute, i.e. 16 GPUs for a single day. Compared to other communities, this is equivalent to medium-scale studies in vision or NLP. We argue that it is finally time for the emergent language community to scale up!

In this spirit, we start clearing the way by identifying some scaling up challenges: optimization instabilities, ill-adapted metrics, or lack of population synergy. We show how to face these new difficulties by proposing: alternative optimization (KL regularization), new evaluation protocols (ETL, zero-shot dataset transfer, best seed baseline), and new dynamics to leverage populations (imitation, voting). There are different theories about the necessity of complex tasks to model human communication (e.g., (Barrett & Skyrms, 2017) vs. (Bickerton, 2015; Dupoux, 2018)). Hence, we endorse here Bickerton's view and adopt performance-inspired solutions to scale up such as KL-regularization and imitation. An interesting future debate is whether our findings could influence the status quo of similar research in human communication.

Although we only examine three scaling up dimensions, many other directions can be pursued in the future: using larger or different architectures to improve agent abilities (Desai & Johnson, 2021), experimenting with multimodal data like video or sounds for more realistic stimuli (Arandjelovic & Zisserman, 2017), or building symmetric communication channels to have emergent dialogues (Gao et al., 2019). Another frontier would be to incorporate interaction within environments to ground language into actions (Bisk et al., 2020). As such embodiment is crucial for human language understanding (Harnad, 1990; Barsalou, 2008), we may start exploring small grid world (Kajić et al., 2020) before scaling up to 3D-environments (Abramson et al., 2020).

## REPRODUCIBILITY STATEMENT

In this paper, we ensure the reproduciblity of the different findings through several (and voluntary redundant) ways, namely:

- The task is detailed in Section 2.1 and we provide a visual sketch in Appendix A.1.
- We use open-source datasets (ImageNet and CelebA). Also, dataset processing is explained, with pseudo-code to reproduce our splits, in Appendix A.6.
- The speaker and listener architectures are first explained in Section 2.2, we then detail model size values, and provide visual sketches in Appendix A.2. The specifics for image reconstruction for *ETL* are in Appendix B.1 with a pseudo-code for the reconstruction head.
- The optimization is first described in Section 2.3. The equations are then fully detailed in Appendix A.3.
- All training and evaluation hyperparameters are listed in Appendix A.4.
- Through the paper, we voluntary provide training curves, test curves, and final scores to have a global view of the training trends.
- We listed computation time and memory footprint for multiple experiments and multiple hardware in Appendix B.2
- The source code should be released upon paper acceptance.

## ACKNOWLEDGEMENT

We would like to thank Will Dabney, Remi Munos, Karl Tuyls, Nathalie Beauguerlange as well as the rest of the DeepMind Paris team for their continuous support. We would also like to thank Marco Baroni, Eugene Kharitonov, Olivier Pietquin and Mathieu Rita for their discussions and helpful feedback at the different stages of the project. Finally, we thank Alison Reid and Saran Tunyasuvunakool for their help in open-sourcing the codebase.

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

# A ADDITIONAL SETTING DETAILS

## A.1 DISCRIMINATION GAME

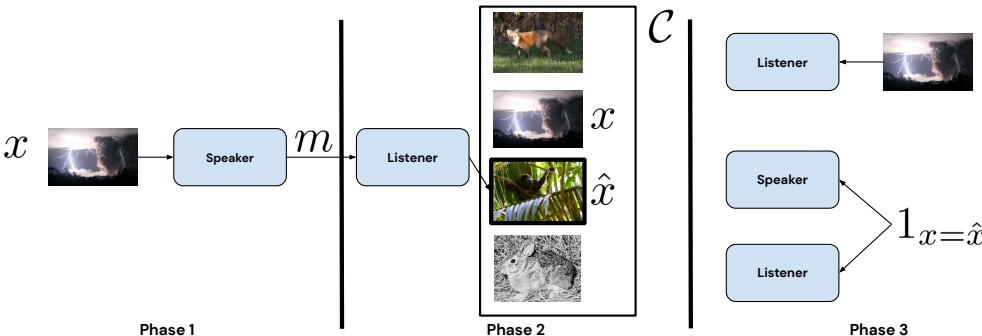

Figure 5: Example of a discrimination game on Imagenet with a set $\mathcal{C}$ of $4$ candidates. In this specific instance, Listener does not select an image $\hat{x}$ that is identical to the target image $x$ received by Speaker. Therefore the reward received by both player $R(x, \hat{x}) = 1_{x=\hat{x}}$ will be 0.

## A.2 ARCHITECTURE DETAILS

**Speaker** The speaker's network architecture is composed of several components to transform the target image $x$ into a message $m = (w_t)_{t=0}^{T-1}$:

- The encoder $f$ is a fixed Resnet-50 architecture that has been previously trained on Imagenet with the BYOL algorithm. The resulting embedding $f(x)$ is of size 2048.

- The RNN $h_\theta$ used is an LSTM of hidden size 256. Therefore the core state $z_{t,\theta}$ is of size 512.

- The core-state adapter $c_\theta$ is a linear layer with input size 2048 and an output size of 512 that allows to transform the embedding $f(x)$ into an appropriate core state $z_{-1,\theta} = c_\theta(f(x))$. We split $z_{-1,\theta}$ into two equal parts to obtain the initial hidden state $z_{h,-1,\theta}$ and the initial cell state $z_{c,-1,\theta}$.

- The word embedder $g_\theta$ associates to each discrete symbols in $\mathcal{W} \cup \{\texttt{sos}\}$ an embedding of size 10.

- The value head $v_\theta$ first selects the hidden part $z_{h,t,\theta}$ of the core state $z_{t,\theta}$ and then applies a linear layer of output size 1.

- The policy head $\pi_\theta$ first selects the hidden part $z_{h,t,\theta}$ of the core state $z_{t,\theta}$ and then applies a linear layer of output size $|\mathcal{W}|$ to obtain logits $l_{t,\theta}$. Then a softmax layer transforms the logits into the output distribution $\pi_\theta(. | z_{t,\theta})$.

**Listener** The listener's network architecture is composed of several components to transform the message $m$ and the input image $\tilde{x}$ into a score $\texttt{score}(m, \tilde{x}, \phi)$:

- The encoder $f$ is a fixed Resnet-50 architecture that has been previously trained on Imagenet with the BYOL algorithm. The resulting embedding $f(x)$ is of size 2048.

- The RNN $h_\phi$ used is an LSTM of hidden size 512. Therefore the core state $z_{t,\phi}$ is of size 1024 and is composed of an hidden state $z_{h,t,\phi}$ of size 512 and a cell state $z_{c,t,\phi}$ of size 512.

- The word embedder $g_\phi$ associates to each discrete symbols in $\mathcal{W}$ an embedding of size 10.

- The target projection $t_\phi$ is a linear layer with output size 256.

- The core-state projection $p_\phi$ is a linear layer with output size 256.

As explained in Sec.2.2, the score function is defined as $\texttt{score}(m, \tilde{x}, \phi) = \cos(\frac{p_{m,\phi}}{\|p_{m,\phi}\|_2}, \frac{t_{\tilde{x}}}{\|t_{\tilde{x}}\|_2})$. The scores over all images are normalized via a softmax to get a probability $\pi_\phi(.|m, \mathcal{C})$ such that:

$$\forall \tilde{x} \in \mathcal{C}, \pi_\phi(\tilde{x}|m, \mathcal{C}) = \frac{\exp(\texttt{score}(m, \tilde{x}, \phi))}{\sum_{\overline{x} \in \mathcal{C}} \exp(\texttt{score}(m, \overline{x}, \phi))}.$$

Finally, the listener selects an image by taking the best guess according to $\pi_\phi$, i.e. $\hat{x} \in \arg\max_{\tilde{x} \in \mathcal{C}} \pi_\phi(\tilde{x}|m, \mathcal{C})$.

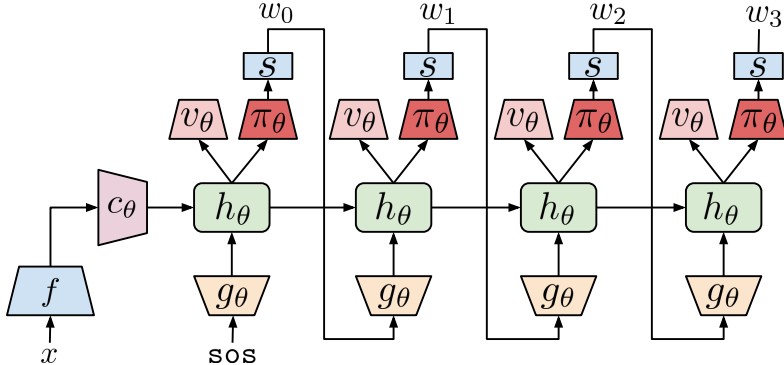

Figure 6: Graphical representation of a speaker's architecture that shows how the words $(w_t)_{t=0}^{T-1}$ are computed from the inputs $x$.

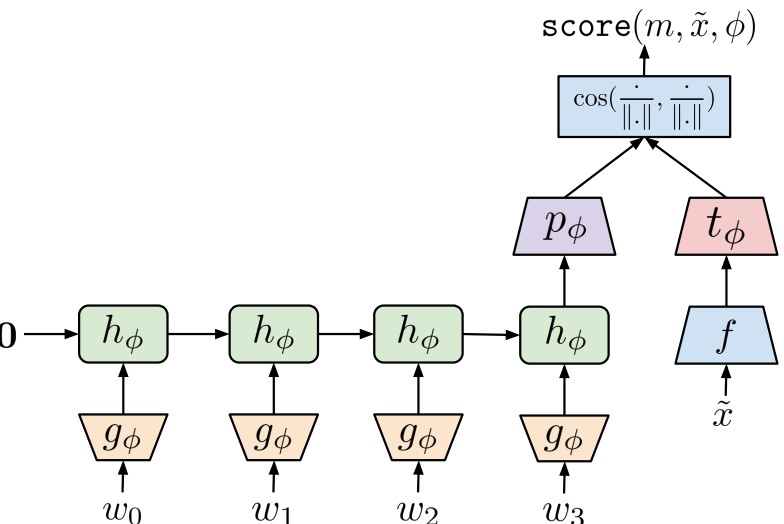

Figure 7: Graphical representation of a listener's architecture that shows how the score $\texttt{score}(m, \tilde{x}, \phi)$ is computed given a message $(w_t)_{t=0}^{T-1}$ and an input image $\tilde{x}$.

### A.3 OPTIMISATION DETAILS

**Speaker training and loss** The goal of a speaker, parameterized by $\theta$, is to optimize the message $m$ sent to a listener, parameterized by $\phi$, such that the mean reward of the game is the highest possible. This can be framed as a sequential decision making problem where the decision is the choice of each word $w_t$. Thus, each word is sampled from a parameterized stochastic policy $\pi_\theta(.|x, (w_k)_{k=0}^{t-1})$ that depends on the image $x$ and past words $(w_k)_{k=0}^{t-1}$, where for $t = 0$ we have $(w_k)_{k=0}^{-1} = \emptyset$. Then, the goal is to maximize the expected reward $J(\theta, \phi)$ by finding the best policy $\pi_\theta$:

$$J(\theta, \phi) = \mathbb{E}_{x \sim \rho}\left[\mathbb{E}_{\pi_\theta, x}[R(x, \hat{x})]\right],$$

where the expectation $\mathbb{E}_{x\sim\rho}$ is over the dataset of training images and the expectation $\mathbb{E}_{\pi_\theta,x}$ is over all possible sequences $(x,(w_t)_{t=0}^{T-1})$ that can be generated by $\pi_\theta$ starting from $x$. For a given image $x$, we define the value $V^{\pi_\theta}(x)=\mathbb{E}_{\pi_\theta,x}[R(x,\hat{x})]$ as the expected reward for a given image, here we left the dependence over the choice of the listener $\hat{x}$ for ease of notations as the speaker cannot act on it. For a parameterized policy $\pi_\theta$, we can use the policy gradient theorem to obtain the gradient:

$$\partial_\theta V^{\pi_\theta}(x)=\mathbb{E}_{\pi_\theta,x}\left[\sum_{t=0}^{T-1}\left(R(x,\hat{x})-V_{t-1}^{\pi_\theta}(x)\right)\partial_\theta\log(\pi_\theta(w_t|x,(w_k)_{k=0}^{t-1}))\right],$$

where $V_{t-1}^{\pi_\theta}(x)=\mathbb{E}_{\pi_\theta,x}\left[R(x,\hat{x})|(w_k)_{k=0}^{t-1}\right]$ is the value conditioned on all the information revealed at time $t-1$. For our particular choice of speaker network, we encode the policy $\pi_\theta(.|x,(w_k)_{k=0}^{t-1})$ and the value $V_{t-1}^{\pi_\theta}(x)$ with the policy head $\pi_\theta(.|z_{t,\theta})$ and the value head $v_\theta(z_{t,\theta})$ respectively. Doing so is legitimate because by construction of our recurrent speaker network the embedding $z_{t,\theta}$ is a function of the image $x$ and the past words $(w_k)_{k=0}^{t-1}$. We train then our speaker network by minimizing a value loss $L_V(\theta)$ to make $v_\theta(z_{t,\theta})$ fit $V_{t-1}^{\pi_\theta}(x)$ over a batch $\mathcal{X}$ of images:

$$L_V(\theta)=\frac{1}{|\mathcal{X}|}\sum_{x\in\mathcal{X}}L_V(\theta,x)=\frac{1}{|\mathcal{X}|}\sum_{x\in\mathcal{X}}\sum_{t=0}^{T-1}\left(R(x,\hat{x})-v_\theta(z_{t,\theta})\right)^2,$$

and a policy loss function $L_\pi(\theta)$ to maximise the expected reward:

$$L_\pi(\theta)=\frac{1}{|\mathcal{X}|}\sum_{x\in\mathcal{X}}L_\pi(\theta,x)=\frac{1}{|\mathcal{X}|}\sum_{x\in\mathcal{X}}\sum_{t=0}^{T-1}\mathtt{sg}\left(R(x,\hat{x})-v_\theta(z_{t,\theta})\right)\log(\pi_\theta(w_t|z_{t,\theta})),$$

where $\mathtt{sg}(.)$ is the stop gradient function. If we assume that $v_\theta(z_{t,\theta})$ fits perfectly $V_{t-1}^{\pi_\theta}(x)$, one can easily verify that $\partial_\theta L_\pi(\theta,x)$ is an unbiased estimate of $-\partial_\theta V^{\pi_\theta}(x)$ therefore $\partial_\theta L_\pi(\theta)$ is also an unbiased estimate of $-\partial_\theta J(\theta,\phi)$. In addition to following the gradient $\partial_\theta V^{\pi_\theta}(x)$, it is common practice in RL and emergent language literature to maximize an entropy term encouraging to explore other choices of words by Speaker:

$$\mathcal{H}(\pi_\theta)=\mathbb{E}_{x\sim\rho}\left[\mathbb{E}_{\pi_\theta,x}\left[\sum_{t=0}^{T-1}\mathcal{H}(\pi_\theta(.|x,(w_k)_{k=0}^{t-1}))\right]\right],$$

$$=\mathbb{E}_{x\sim\rho}\left[\mathbb{E}_{\pi_\theta,x}\left[\sum_{t=0}^{T-1}\mathbb{E}_{w\sim\pi_\theta(.|x,(w_k)_{k=0}^{t-1})}\left[\pi_\theta(w|x,(w_k)_{k=0}^{t-1})\log(\pi_\theta(w|x,(w_k)_{k=0}^{t-1}))\right]\right]\right].$$

In practice, a sampled version of the negative entropy $L_\mathcal{H}(\theta)$ is minimized with the speaker network:

$$L_\mathcal{H}(\theta)=-\frac{1}{|\mathcal{X}|}\sum_{x\in\mathcal{X}}\sum_{t=0}^{T-1}\mathbb{E}_{w\sim\pi_\theta(.|z_{t,\theta})}\left[\pi_\theta(w|z_{t,\theta})\log(\pi_\theta(w|z_{t,\theta}))\right].$$

Recently, several deep RL (Schulman et al., 2015; 2017) and theoretical RL papers (Geist et al., 2019; Vieillard et al., 2020a;b) argued that minimizing the KL between the online policy $\pi_\theta$ and a target policy $\pi_{\bar{\theta}}$ instead of or in addition to entropy regularization could be beneficial for better final performance as well as stabilizing the learning. Generally, $\pi_{\bar{\theta}}$ is an older policy or an exponential moving average of past policies. Therefore, we also consider the following KL regularization $\mathtt{KL}(\pi_\theta,\pi_{\bar{\theta}})$:

$$\mathtt{KL}(\pi_\theta,\pi_{\bar{\theta}})=\mathbb{E}_{x\sim\rho}\left[\mathbb{E}_{\pi_\theta,x}\left[\sum_{t=0}^{T-1}\mathtt{KL}(\pi_\theta(.|x,(w_k)_{k=0}^{t-1}),\pi_{\bar{\theta}}(.|x,(w_k)_{k=0}^{t-1}))\right]\right],$$

$$=\mathbb{E}_{x\sim\rho}\left[\mathbb{E}_{\pi_\theta,x}\left[\sum_{t=0}^{T-1}\mathbb{E}_{w\sim\pi_\theta(.|x,(w_k)_{k=0}^{t-1})}\left[\pi_\theta(w|x,(w_k)_{k=0}^{t-1})\log(\frac{\pi_\theta(w|x,(w_k)_{k=0}^{t-1})}{\pi_{\bar{\theta}}(w|x,(w_k)_{k=0}^{t-1})})\right]\right]\right].$$

In practice, the policy $\pi_{\bar{\theta}}$ is obtained by doing an exponential moving average of the weights $\theta$ over training. Then, a sampled version of the KL is minimized $L_{\mathtt{KL}}(\theta)$ with our specific speaker network:

$$L_{\mathtt{KL}}(\theta)=\frac{1}{|\mathcal{X}|}\sum_{x\in\mathcal{X}}\sum_{t=0}^{T-1}\mathbb{E}_{w\sim\pi_\theta(.|z_{t,\theta})}\left[\pi_\theta(w|z_{t,\theta})\log(\frac{\pi_\theta(w|z_{t,\theta})}{\pi_{\bar{\theta}}(w|z_{t,\bar{\theta}})})\right].$$

To sum up, the speaker training loss $L(\theta)$ on a batch of images $\mathcal{X}$ is:

$$L(\theta) = L_V(\theta) + L_\pi(\theta) + \alpha L_\mathcal{H}(\theta) + \beta L_{\mathrm{KL}}(\theta),$$

where $\alpha$ and $\beta$ are hyper-parameters.

**Listener loss**   One important detail in Sec. 2.3 is that for each image $x$ in a batch $\mathcal{X}$, a set of image candidates $\mathcal{C}(x, \mathcal{X})$ is sampled randomly (uniform without replacement over $\mathcal{X} \setminus \{x\}$) chosen in $\mathcal{X}$. We omitted the dependencies of $\mathcal{C}(x, \mathcal{X})$ on $x$ and $\mathcal{X}$ in the main text for ease of reading. In addition, as explained in Sec. 2.3, the listener loss is a multi-class classification loss where the correct class is the index of $x$ in the set of candidates $\mathcal{C}$ also called InfoNCE loss (van den Oord et al., 2018): $L(\phi) = -\frac{1}{|\mathcal{X}|} \sum_{x \in \mathcal{X}} \log (\pi_\phi(x|m, \mathcal{C}))$. This can be rewritten more explicitly:

$$
\begin{aligned}
L(\phi) &= -\frac{1}{|\mathcal{X}|} \sum_{x \in \mathcal{X}} \log (\pi_\phi(x|m, \mathcal{C})), \\
&= -\frac{1}{|\mathcal{X}|} \sum_{x \in \mathcal{X}} \log \left( \frac{\exp(\texttt{score}(m, x, \phi))}{\sum_{\overline{x} \in \mathcal{C}} \exp(\texttt{score}(m, \overline{x}, \phi))} \right), \\
&= -\frac{1}{|\mathcal{X}|} \sum_{x \in \mathcal{X}} \log \left( \frac{\exp(\cos(\frac{p_{m,\phi}}{\|p_{m,\phi}\|_2}, \frac{t_x}{\|t_x\|_2}))}{\sum_{\overline{x} \in \mathcal{C}} \exp(\cos(\frac{p_{m,\phi}}{\|p_{m,\phi}\|_2}, \frac{t_{\overline{x}}}{\|t_{\overline{x}}\|_2}))} \right).
\end{aligned}
$$

**Imitation training among a group of speakers.**   In a training imitation step, a group of $K$ speakers among the total population of $N$ speakers is selected. Among those $K$ speakers, one speaker is going to play the role of the teacher and $K - 1$ are going to play the role of the students. To choose the teacher among the $K$ speakers, we use as metric the exponential moving average of the accuracies over each batch on which a given speaker has been trained on. To be more precise, let $\theta_i$ be the $i$-th speaker and $\mathcal{X}$ the present batch on which $\theta_i$ has just been trained on, then $\sigma_i$ is updated according to the following rule:

$$\sigma_i \leftarrow \mu \sigma_i + (1 - \mu) \frac{1}{|\mathcal{X}|} \sum_{x \in \mathcal{X}} 1_{x=\hat{x}}$$

and is the exponential moving average of coefficient $\mu$ of the accuracies over each batch on which speaker $i$ has been trained. Then the teacher is simply the speaker with the highest $\sigma_i$ among the $K$ speakers. Now for convenience, let us note $\theta_T$ the parameters of the teacher and $\theta_S$ the parameters of a student. A student $\theta_S$ is going to be trained on a batch of data $\mathcal{X}$ by imitating the messages of the teacher $\theta_T$ with the following loss:

$$L_I(\theta_S) = -\frac{1}{|\mathcal{X}|} \sum_{x \in \mathcal{X}} \sum_{t=0}^{T-1} \log \pi_{\theta_S}(w_t(x, \theta_T)|x, z_{\theta_S, t}).$$

The loss $L_I(\theta_S)$ is a cross-entropy loss that encourages the student to output the same words as the teacher.

**Optimizer hyper-parameters**   Each member of the population (listener or speaker) uses its own optimiser. All optimisers are Adam optimisers (Kingma & Ba, 2015) with the same set of hyper-parameters:

- learning rate: $1e - 4$,
- $\beta_1$: 0.9,
- $\beta_2$: 0.999,
- $\epsilon$: $1e^{-8}$.

For the speakers' loss, we use $\alpha = 0.0001$ for the entropy regularization and $\beta = 0.5$ for the KL regularization. See Sec. B.3 for more details about the impact of these parameters.

## A.4 HYPER-PARAMETERS

We use the same hyper-parameters across the different settings. When experimenting with ImageNet vs. CelebA, we only vary the number of maximum steps. Specifically, we use 300k maximum steps for CelebA (we already start observing some overfitting with this value) and 900k maximum steps for ImageNet. In all cases, we select the best checkpoint for evaluation with respect to the listener loss at validation time. For robust evaluation, we average the scores over 5 epochs to have different candidates for a given target and get a score less dependent on the sampling of the candidates. Unless mentioned otherwise, the remaining training hyper-parameters are reported in Table 3.

Table 3: Hyper-parameters values across datasets and settings.

| | | |
|---|---|---|
| Learning rate | $lr$ | 0.0001 |
| Batch training size | $|\mathcal{X}|$ | 1024 |
| Number of Candidates | $|\mathcal{C}|$ | 1024 |
| Number of agent sampled | $P$ | min(N, 10) |
| KL coefficient | $\beta$ | 0.5 |
| KL EMA | $\eta$ | 0.99 |
| Entropy Coefficient | $\alpha$ | 0.0001 |
| Vocabulary size | $|\mathcal{W}|$ | 20 |
| Message Length | $T$ | 10 |
| Imitation EMA | $\mu$ | 0.99 |
| *ETL* learning rate | *ETL*, $lr$ | 0.001 |
| *ETL* training batch size | *ETL*, $|\mathcal{X}|$ | 4096 |
| *ETL* training candidates | *ETL* discr., $|\mathcal{C}|$ | 4096 |

## A.5 IMITATION HYPER-PARAMETERS

In all our experiments, we perform a grid search over the parameters of $M$ (number of interaction steps) and $K$ (number of sampled speakers at the imitation step). For the case of a population of $N = 10$, $K$ is chosen from $\{1, 4, 9\}$ and $M$ from $\{10, 50, 100\}$ according to the validation accuracy. For $N = 50$, $K$ is selected from $\{4, 9, 24\}$ and $M$ from $\{1, 10, 50\}$. Table 4 shows the selected hyper-parameters $(K, M)$ for the different settings.

Table 4: Imitation hyper-parameters values chosen according to the best accuracy at validation.

| Dataset | 10 pairs + imitation $(K, M)$ | 50 pairs + imitation $(K, M)$ |
|---|---|---|
| CelebA | (1, 10) | (9, 10) |
| ImageNet | (1, 100) | (9, 10) |

A.6    DATASETS DETAILS

**ImageNet**    ILSVRC2012, also known as ImageNet (Deng et al., 2009; Russakovsky et al., 2015) is among the historical largest natural RGB image dataset. It mostly contains images of animals and objects over 1000 labels, e.g., camel, ostrich, hourglass, or bow tie. In our experiments, we use 99% of the official train set for training, i.e., 1300k images, the last 1% of the train set for validation, i.e., 13k images, and the official validation set as our test set (i.e., 50k images).

**CelebA**    CelebA is a large natural RGB dataset (Liu et al., 2015), which contains image of celebrity faces over 10,177 identities. Noticeably, each face also contains 40 binary attributes, e.g., glasses and hair. Such attributes are interesting to compute, for instance, language topography. Although there is an official CelebA split, there are no overlapping identities between the train, validation, and test set. We thus reshuffle the split as described in Listing 1. In the end, we respectively obtain 169,444 training, 16,669 valid, and 16,486 test images.

**Image Processing**    In both datasets, we resize images to 256 pixels along the shorter side using bicubic resampling before applying a 224x224 center crop. We normalize the color channels subtracting the mean color and dividing by ImageNet std. We then use the ResNet-50 encoder pre-trained on ImageNet with the self-supervised method BYOL (Grill et al., 2020) to extract the final representation of dimension 2048.

Listing 1: CelebA Splits

```python
SPLIT_RATIO = 5   # the ratio train:valid is 5:1

def celeba_splits(data):
    """Split the data in three sets with overlapping identities"""

    # Initialize variables
    image_train, image_valid, image_test = [], [], []
    counter_label = collections.Counter()

    for data in dataset.values():

        label = data['label']   # label encode the face identity
        count = counter_label[label]

        if count > 0 and count % SPLIT_RATIO == 0:
            # We equally split valid and test by Even and Odd labels
            if label % 2 == 0:
                image_valid.append(data)
            else:
                image_test.append(data)
        else:
            image_train.append(data)

        counter_label[label] += 1

    return image_train, image_valid, image_test
```

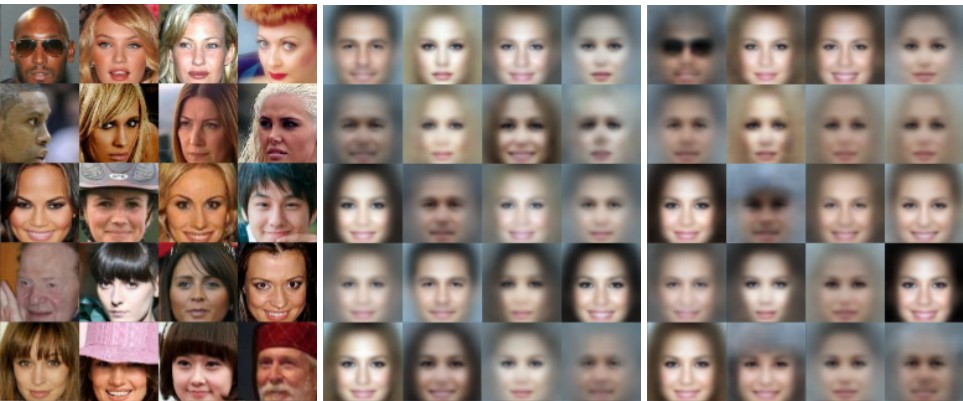

Figure 8: Face reconstructions. Left: Randomly sampled input images from the CelebA dataset; Middle: Reconstructions, using messages from a model trained with 16 candidates; Right: Reconstructions, using messages from a model trained with 1024 candidates.

## B    ADDITIONAL RESULTS AND ANALYSIS

### B.1    FACE RECONSTRUCTION

To visualize some of the features encapsulated in the messages used by the agent, we propose a face reconstruction procedure. For a single speaker, parameterized by $\theta$, and given an initial image $x \in \mathcal{X}$, we produce a message $m = (w_t)_{t=0}^{T-1}$. We feed the messages to a listener-like architecture, consisting of an embedding layer of size 10 followed by an LSTM with 512 units. We keep the last output of the LSTM, $h_{T-1}$ as a message representation. We then pass the message through a deconvolutional architecture, similar to the one in Radford et al. (2016), but without batch normalizations. Pseudocode for the architecture is provided in Listing 2. The full network is trained by minimizing the $\ell^2$ loss between the input image and the reconstructed one. To optimize the loss, we use AdamW (Kingma & Ba, 2015) with a batch size of 128, a learning rate of $3 \cdot 10^{-4}$, $\beta_1 = .9$, $\beta_2 = .9$, $\varepsilon = 10^{-8}$ and a weight decay of 0.01. We clip gradients by norm with a clipping threshold of 500, and skip gradient updates with norm superior to 2000. In Table 1, we report reconstruction losses for different number of candidates $|\mathcal{C}|$. The reconstruction loss is computed as a squared error, summed over both spatial and channel axes. Figure 8 displays examples of input images, and corresponding reconstructions for two discrimination games with different number of candidates.

Optimally, each reconstruction should converge to the average face, given the corresponding message. The more features a message contains, the better reconstructions should be. Note first that reconstructions are far from perfect, and much worse than reconstructions that an auto-encoder trained end to end on images would provide: first, solving the discrimination game does not require fully reconstructing the input image, but only capturing enough features to identify uniquely the image in a batch of candidates; second, the discrete nature of the message, its limited size, and the fact that it is learnt using reinforcement learning act as strong bottlenecks, that prevents messages from containing all the information about input images.

Nonetheless, as qualitatively shown in Figure 8, messages contain semantic information about inputs images. For instance, hair color, gender, or open mouth are mostly preserved throughout the reconstruction process. Other features, such as face orientation, skin tone, or age are mostly ignored in messages. Furthermore, both quantitative and qualitative difference are visible when going from low number of discriminators to high number of candidates. Quantitatively, going from 16 to 1024 candidates improves the reconstruction error from $2448_{\pm 16}$ to $2351_{\pm 14}$ (Table 1). Qualitatively, messages with 1024 candidates seem to contain information about presence of eye-glasses (top left image), as well as presence of head cover (bottom right image), that are absent in messages from the Speaker trained with 16 candidates.

Listing 2: Face reconstruction Head

```python
def upsample_2d(x, factor: int = 2):
  bs, height, width, channels = x.shape
  x = image.resize(
      x, (bs, height * factor, width * factor, channels), method='nearest')
  return x

def img_reconstruction(x):
    """"Take the LSTM output, and turn it into an image."""

    # Project the flat embedding into a 2d tensor of dim 4x4x128
    x = nn.Linear(4 * 4 * 128)(x)
    x = x.reshape((x.shape[0], 4, 4, 128))
    x = nn.relu(x)

    x = upsample_2d(x)   # 8x8
    x = nn.Conv3x3(64, with_bias=False, padding="VALID")(x)
    x = nn.relu(x)

    x = upsample_2d(x)   # 16x16
    x = nn.Conv3x3(32, with_bias=False, padding="VALID")(x)
    x = nn.relu(x)

    x = upsample_2d(x)   # 32x32
    x = nn.Conv3x3(16, with_bias=False, padding="VALID")(x)
    x = nn.relu(x)

    x = upsample_2d(x)   # 64x64
    x = nn.Conv3x3(16, with_bias=False, padding="VALID")(x)
    x = nn.relu(x)
    x = nn.Conv3x3(3, with_bias=False, padding="VALID")(x)

    return nn.tanh(x)
```

## B.2 COMPUTATIONAL REQUIREMENTS

As mentioned in the article, our approach remains computationally tractable with widely available hardware. Table 5 summarizes these requirements to reproduce our experimental setup.

## B.3 IMPACT OF THE KL REGULARIZATION ON TRAINING STABILITY

We showed in the main paper how training a pair of agents to solve a complex task (a discrimination game with 1024 candidates) becomes unstable when using common optimization algorithms. In this section, we look at the training curves when agents are trained with and without KL regularization, while varying the entropy coefficient $\alpha \in \{10^{-4}, 10^{-3}, 10^{-2}, 10^{-1}\}$. Results are shown with a population of size 1 (Figure 9) and 10 (Figure 10). In both cases, agents are trained to solve the complex discrimination task with 1024 candidates.

First, we observe that the lower $\alpha$ is, the more crucial applying a KL regularization is. For example, in Figure 9(a), we go from a chaotic setting that converges to an accuracy of $\sim 40\%$ when the KL coefficient $\beta = 0$ to a significantly more stable optimization with almost perfect accuracy when $\beta \geq 0.5$. Second, we observe that, if the KL regularization is useful for stable optimization, it comes at the cost of the rapidity of convergence. This is clearly shown in Figures 9(c) and 10(c), where we observe that the larger $\beta$ is, the slower the convergence is. In other words, one should select the best $\beta$ optimizing the stability/rapidity trade-off. Third, in the presence of a KL regularization, training stability depends less on the entropy coefficient $\alpha$. Indeed, we observe a stable and high training

Table 5: Computational requirements for our base setup. "GPU memory" refers to the peak GPU memory usage.

| Dataset | Device | Pop. size, $N$ | GPU memory (GiB) | Step time (ms) | Train. time (hours) |
|---|---|---|---|---|---|
| ImageNet | p100 | 1 | 0.29 | 42 | 10.5 |
| | | 10 | 0.71 | 381 | 95.2 |
| | | 50 | 2.63 | 1887 | 471.7 |
| | v100 | 1 | 0.29 | 25 | 6.3 |
| | | 10 | 0.71 | 223 | 55.7 |
| | | 50 | 2.63 | 1089 | 272.1 |
| CelebA | p100 | 1 | 0.33 | 89 | 7.4 |
| | | 10 | 0.75 | 457 | 38.1 |
| | | 50 | 2.60 | 2248 | 187.4 |
| | v100 | 1 | 0.33 | 107 | 8.9 |
| | | 10 | 0.75 | 293 | 24.4 |
| | | 50 | 2.60 | 1408 | 117.3 |

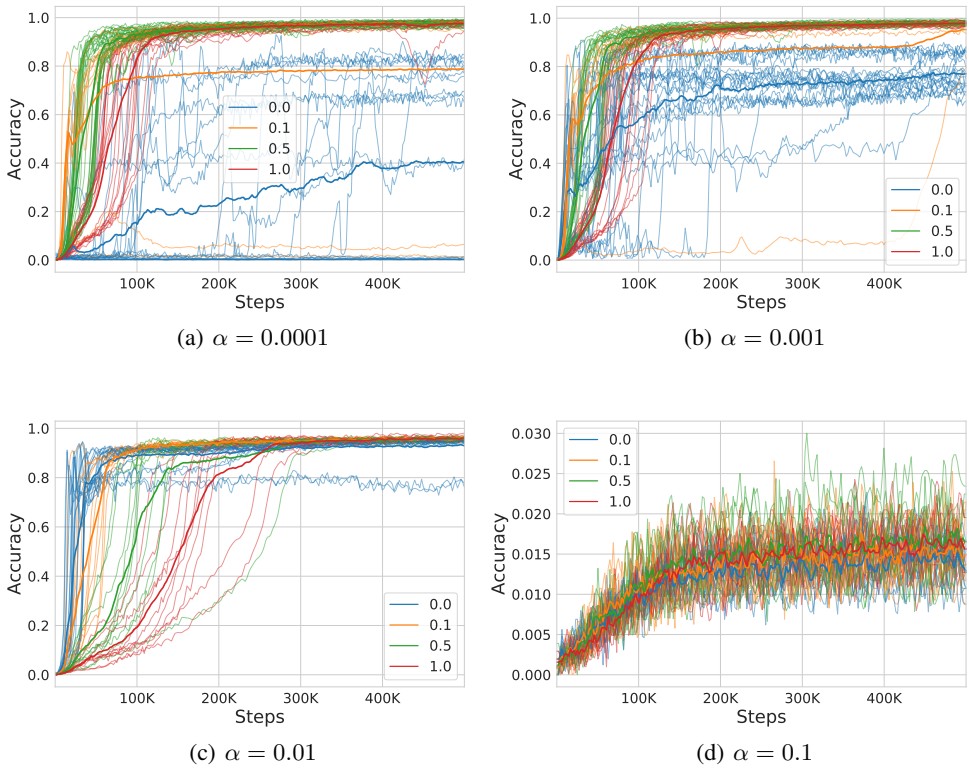

(a) $\alpha = 0.0001$  (b) $\alpha = 0.001$

(c) $\alpha = 0.01$  (d) $\alpha = 0.1$

Figure 9: Training accuracy for a 1 communicating pair when varying the KL coefficient $\beta$. Each sub-figure represents the results for a fixed entropy coefficient $\alpha$. Thin lines represent the training accuracy curves of different seeds. Thick lines represent the average across 10 seeds.

curves for $\alpha \in \{10^{-4}, 10^{-3}, 10^{-2}\}$.[4] However, training without KL regularization (blue curves with $\beta = 0$) is very sensitive to the value of $\alpha$ and the most stable case is observed for $\alpha = 0.01$. Still even in this best case without KL regularization, having a $\beta > 0$ is useful as shown in Figure 11.

---

[4] It is also the case for $\alpha = 0$. Not shown here.

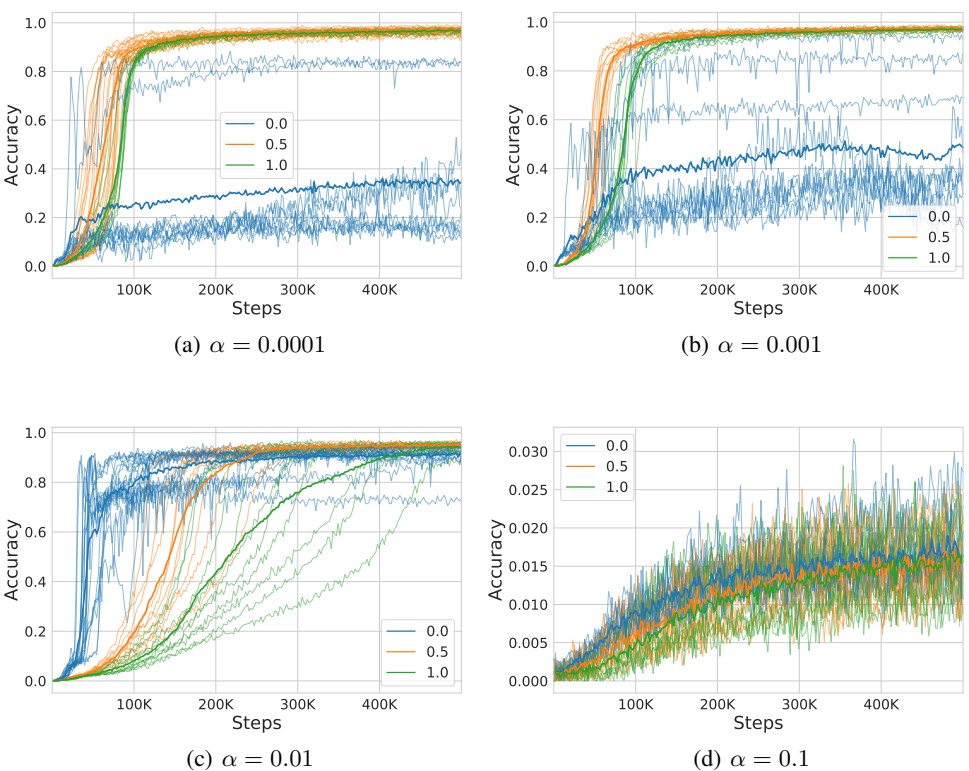

Figure 10: Training accuracy for a 10 communicating pairs when varying the KL coefficient $\beta$. Each sub-figure represents the results for a fixed entropy coefficient $\alpha$. Thin lines represent the training accuracy curves of different seeds. Thick lines represent the average across 10 seeds.

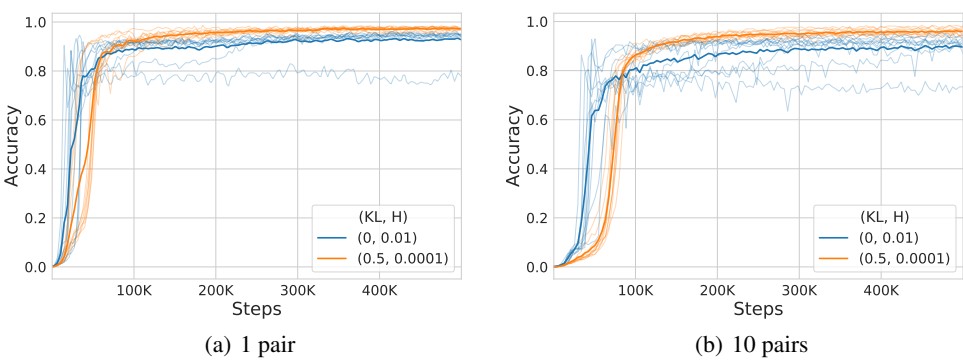

Figure 11: Training accuracy for 1 and 10 pairs. Each sub-figure compares the best setting with no KL regularization (with entropy coefficient of 0.01) and the selected setting for our experiments, with a KL coefficient of 0.5 and entropy coefficient of 0.0001. Thin lines represent the training accuracy curves of different seeds. Thick lines represent the average across 10 seeds. In both cases, the setting with the KL regularization, $(0.5, 0.0001)$, exhibits a more stable convergence with a larger difference for population of 10 pairs.

Table 6: Different language properties on CelebA dataset, in %. For each setting we report the mean over 10 seeds. $\pm$ denotes 1 standard error of the mean.

| Setting | Generalization | Robustness |
|---|---|---|
| best 1 pair | 90.73 | 35.82 |
| 1 pair | $89.00_{\pm 0.48}$ | 37.90 |
| 10 pairs | $91.06_{\pm 0.23}$ | 37.56 |
| 50 pairs | $90.69_{\pm 0.61}$ | 38.87 |
| 10 pairs + imitation | $91.84_{\pm 0.31}$ | 35.47 |
| 10 pairs + imitation + vote | $92.19_{\pm 0.30}$ | 35.21 |
| 50 pairs + imitation | $92.82_{\pm 0.51}$ | 32.69 |
| 50 pairs + imitation + vote | $\mathbf{93.13}_{\pm \mathbf{0.50}}$ | 32.40 |
| 1 pair + reset | $92.21_{\pm 0.44}$ | **18.34** |
| 10 pairs + reset | $90.55_{\pm 0.56}$ | 35.32 |
| 50 pairs + reset | $91.52_{\pm 0.42}$ | 34.46 |

### B.4 STUDY OF THE RESETTING ON THE CELEBA DATASET

In this work, we argue that training deep agents to communicate in a population benefits from exploiting its richness, as opposed to a standard training (Mordatch & Abbeel, 2018). In this context, we introduced imitation and voting mechanisms. However, other prior works have stipulated a similar argument by focusing instead on the idea of the expressivity/compressibility trade-off (Kirby et al., 2014). The latter attests that our language is shaped by the two competing pressures of expressivity and compressibility. In practice, there were different methods to implement this trade-off in deep agents communication such as iterated learning (Ren et al., 2019), cultural transmission (Cogswell et al., 2019), or ease-of-teaching (Li & Bowling, 2019). In this section, we compare imitation and voting to the ease-of-teaching.[5] To encourage languages' ease-of-teaching, Li & Bowling (2019) trained deep agents to solve a discrimination task, while periodically resetting listeners. In this section, we study the impact of this baseline on the different emergent languages' properties introduced in the main paper. Specifically, we consider 3 additional settings: (1) "1 pair + reset" that consists of resetting the only listener, (2) "10 pairs + reset" where we reset a randomly selected listener among the 10 available ones, and (3) "50 pairs + reset" where we reset a randomly selected listener among the 50 available ones. In all cases, resetting takes place every 51k steps. Note that if (1) is identical to Li & Bowling (2019) setting, (2) and (3) present 10 and 50 speakers respectively unlike 1 speaker as it is the case in (Li & Bowling, 2019).

We observe in Table 6 that resetting has a noticeable benefit only in the case of "1 pair" on the generalization and robustness of the languages. In particular, we note an average *Generalization* of 89.00% for "1 pair" vs. 92.21% for "1 pair + reset" and an average *Robustness* of 18.34% vs. 35.82%. However, there is no systematic improvement when agents are trained in a population, ("10 pairs" vs. "10 pairs + resetting") and ("50 pairs" vs. "50 pairs + resetting"). This is in line with Li & Bowling (2019)'s results which show that an abrupt change during training leads to better results. Still, "1 pair + reset" does not systematically improve the emergent languages' properties compared to the population when imitation and voting are at play.

However, as mentioned in the original paper, the purpose of resetting is not to boost agents' generalization or robustness, but instead to incentivize agents to develop easy-to-transmit languages. In the following, we look at *ETL* shown in Figure 12. Here, adding resetting does not lead to a significant change showing curves almost identical to the ones with standard training in all cases.

Overall, our results suggest that resetting listeners is only beneficial in the one-pair agents in some cases. Furthermore, resetting does not induce faster or better easy-to-learn communication protocols over transfer tasks compared to the standard training of the Lewis game (with or without population).

---

[5]We also considered the setup of Cogswell et al. (2019). However, preliminary results showed systematically worse results compared to the ease-of-teaching settings. Our codebase provides both options.

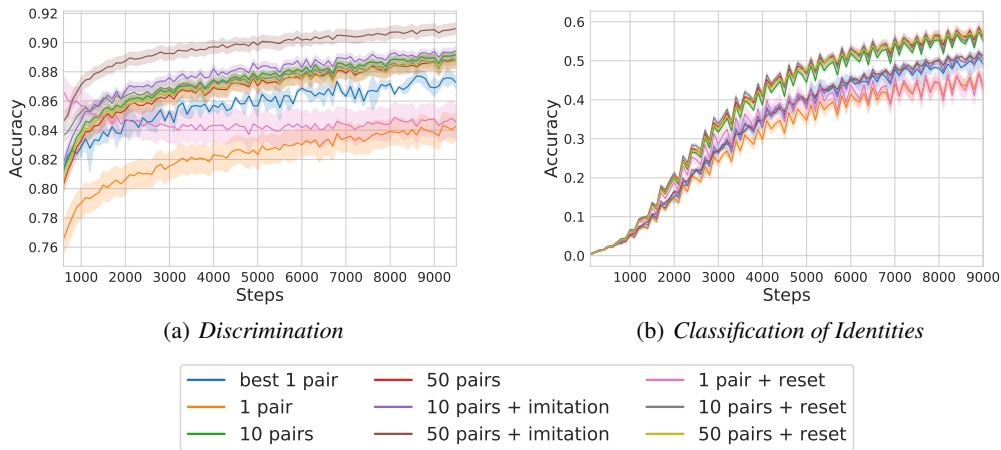

(a) *Discrimination*  (b) *Classification of Identities*

Figure 12: *ETL* for the CelebA dataset of the emergent languages for different tasks. The results are averaged across the emergent languages of $min(5, N)$ different Speakers, newborn listeners' seeds, and across the 10 seeds of each setting. The shaded region represents the standard deviation.

## B.5 IMPACT OF THE NUMBER OF CANDIDATES AT TRAINING AND EVALUATION TIME

In this part we look the impact of task complexity at train (Figure 14) and evaluation (Figure 13) times, by varying the number of candidates $|\mathcal{C}|$. The results confirm our findings in the main paper. That is, for a fixed train $|\mathcal{C}|$, the harder the evaluation task is, the lower the test accuracy is. Still, if the task is complex enough at train time (e.g., $|\mathcal{C}| = 1024$), agents reach overall higher accuracies for all evaluation settings, confirming the importance of hard training tasks to see the emergence of communication protocols with good generalization performances. Also, similarly to our results in the main paper, we observe in Figure 14 that harder evaluation task is necessary to discriminate between the communication protocols.

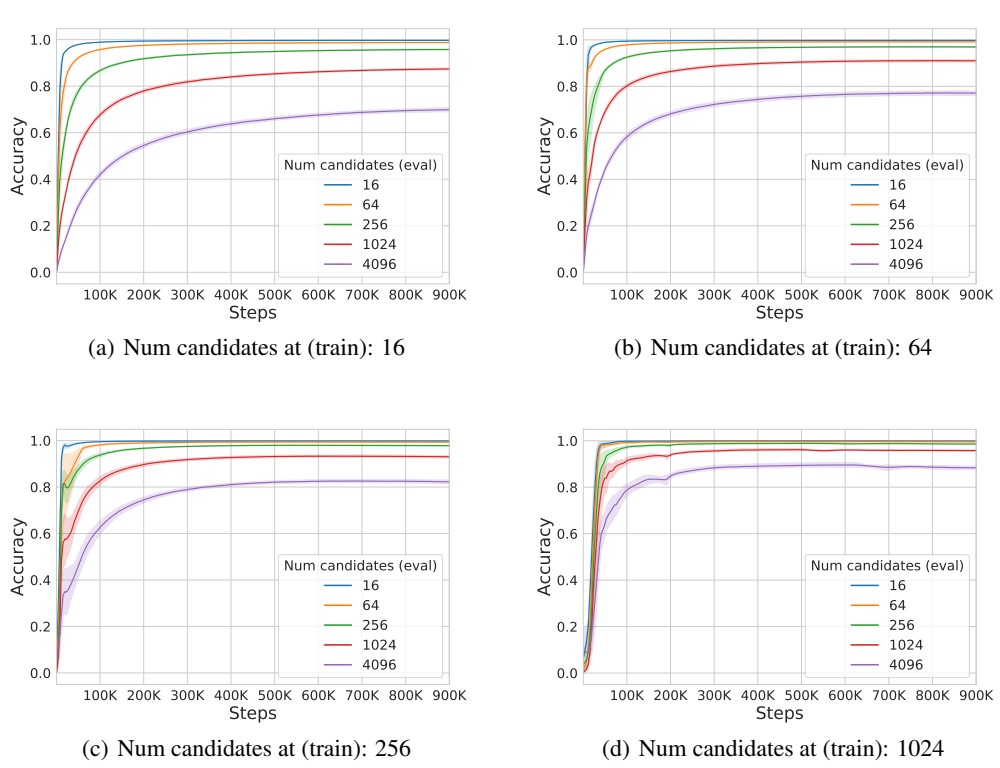

Figure 13: Test accuracy for different number of candidates at training time (subplot) and at evaluation time (lines) on ImageNet. As one would expect, given one model trained with $|\mathcal{C}|$ candidates, the more complex the evaluation task, the lower is the accuracy.

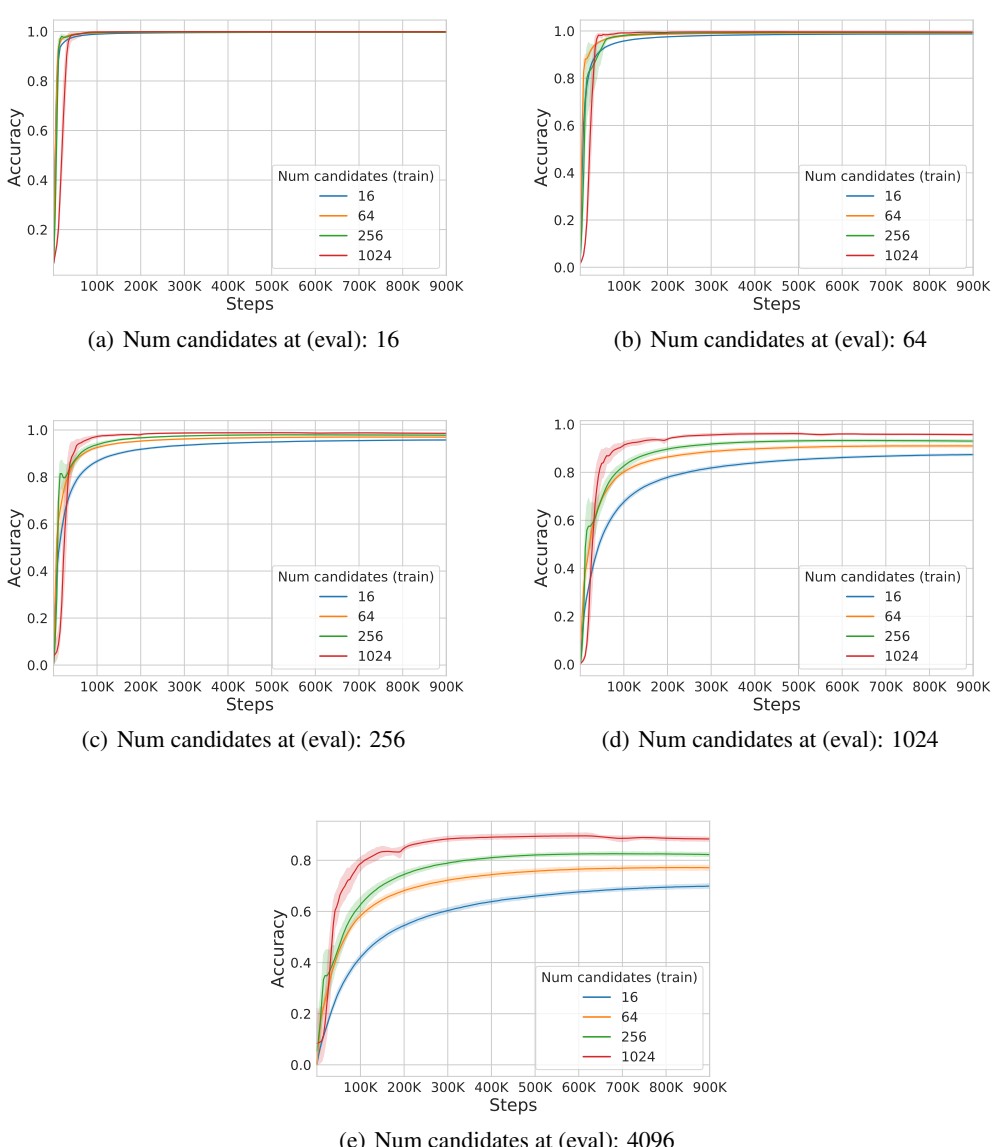

Figure 14: Test accuracy for different number of candidates at training time (lines) and at evaluation time on ImageNet. The more we complexify evaluation, the better we discriminate representations.

## B.6 CELEBA ATTRIBUTES:

Table 7: CelebA *ETL* attribute accuracies when varying the number of candidates at pretraining at 10k training steps.

| Atribute | $|\mathcal{C}| = 16$ | $|\mathcal{C}| = 64$ | $|\mathcal{C}| = 256$ | $|\mathcal{C}| = 1024$ |
|---|---|---|---|---|
| 5 o Clock Shadow | $89.93_{\pm 0.74}$ | $90.57_{\pm 0.75}$ | $91.46_{\pm 0.5}$ | $92.76_{\pm 0.54}$ |
| Arched Eyebrows | $77.73_{\pm 0.75}$ | $79.49_{\pm 0.88}$ | $81.35_{\pm 1.15}$ | $83.22_{\pm 0.8}$ |
| Attractive | $78.74_{\pm 1.01}$ | $80.08_{\pm 0.96}$ | $81.25_{\pm 0.75}$ | $83.27_{\pm 1.18}$ |
| Bags Under Eyes | $81.86_{\pm 0.85}$ | $83.53_{\pm 0.72}$ | $84.66_{\pm 0.68}$ | $86.51_{\pm 0.77}$ |
| Bald | $98.03_{\pm 0.15}$ | $98.03_{\pm 0.23}$ | $98.20_{\pm 0.25}$ | $98.43_{\pm 0.21}$ |
| Bangs | $85.60_{\pm 0.53}$ | $86.29_{\pm 0.71}$ | $87.05_{\pm 0.71}$ | $88.42_{\pm 0.62}$ |
| Big Lips | $77.57_{\pm 0.92}$ | $78.73_{\pm 0.81}$ | $80.31_{\pm 0.97}$ | $82.09_{\pm 0.87}$ |
| Big Nose | $80.58_{\pm 0.74}$ | $82.56_{\pm 0.77}$ | $83.77_{\pm 0.77}$ | $85.56_{\pm 0.75}$ |
| Black Hair | $78.22_{\pm 0.83}$ | $79.78_{\pm 1.32}$ | $81.32_{\pm 1.39}$ | $83.52_{\pm 0.93}$ |
| Blond Hair | $88.23_{\pm 2.31}$ | $89.42_{\pm 1.77}$ | $90.62_{\pm 2.07}$ | $91.94_{\pm 1.01}$ |
| Blurry | $95.11_{\pm 0.19}$ | $95.40_{\pm 0.26}$ | $95.65_{\pm 0.28}$ | $96.31_{\pm 0.41}$ |
| Brown Hair | $80.83_{\pm 0.81}$ | $81.82_{\pm 0.81}$ | $83.26_{\pm 1.0}$ | $84.80_{\pm 0.75}$ |
| Bushy Eyebrows | $86.53_{\pm 0.5}$ | $87.28_{\pm 0.51}$ | $87.87_{\pm 0.54}$ | $89.46_{\pm 0.5}$ |
| Chubby | $94.53_{\pm 0.34}$ | $94.89_{\pm 0.35}$ | $95.44_{\pm 0.34}$ | $96.06_{\pm 0.36}$ |
| Double Chin | $95.76_{\pm 0.34}$ | $95.97_{\pm 0.35}$ | $96.37_{\pm 0.25}$ | $96.86_{\pm 0.26}$ |
| Eyeglasses | $94.50_{\pm 0.53}$ | $95.98_{\pm 1.5}$ | $98.30_{\pm 0.95}$ | $98.48_{\pm 0.76}$ |
| Goatee | $94.28_{\pm 0.38}$ | $94.66_{\pm 0.36}$ | $95.00_{\pm 0.37}$ | $95.88_{\pm 0.44}$ |
| Gray Hair | $96.03_{\pm 0.2}$ | $96.45_{\pm 0.3}$ | $96.81_{\pm 0.25}$ | $97.41_{\pm 0.25}$ |
| Heavy Makeup | $88.08_{\pm 0.5}$ | $88.75_{\pm 0.64}$ | $89.36_{\pm 0.65}$ | $90.49_{\pm 0.95}$ |
| High Cheekbones | $77.94_{\pm 1.87}$ | $79.70_{\pm 1.07}$ | $79.98_{\pm 1.57}$ | $82.39_{\pm 1.33}$ |
| Male | $93.15_{\pm 0.44}$ | $93.82_{\pm 0.43}$ | $94.06_{\pm 0.51}$ | $94.73_{\pm 0.57}$ |
| Mouth Slightly Open | $71.25_{\pm 2.1}$ | $73.62_{\pm 1.2}$ | $74.77_{\pm 1.98}$ | $77.86_{\pm 1.53}$ |
| Mustache | $95.91_{\pm 0.32}$ | $96.15_{\pm 0.38}$ | $96.54_{\pm 0.33}$ | $97.06_{\pm 0.28}$ |
| Narrow Eyes | $89.12_{\pm 0.49}$ | $89.53_{\pm 0.55}$ | $89.98_{\pm 0.53}$ | $90.35_{\pm 0.45}$ |
| No Beard | $88.48_{\pm 0.68}$ | $89.05_{\pm 0.78}$ | $90.21_{\pm 0.84}$ | $91.82_{\pm 0.89}$ |
| Oval Face | $74.54_{\pm 0.76}$ | $76.21_{\pm 0.82}$ | $78.15_{\pm 0.9}$ | $80.82_{\pm 0.9}$ |
| Pale Skin | $95.69_{\pm 0.23}$ | $95.94_{\pm 0.24}$ | $96.13_{\pm 0.31}$ | $96.50_{\pm 0.31}$ |
| Pointy Nose | $75.61_{\pm 0.84}$ | $77.31_{\pm 1.03}$ | $78.78_{\pm 1.23}$ | $80.96_{\pm 0.79}$ |
| Receding Hairline | $92.49_{\pm 0.39}$ | $92.88_{\pm 0.3}$ | $93.23_{\pm 0.39}$ | $93.95_{\pm 0.42}$ |
| Rosy Cheeks | $93.79_{\pm 0.24}$ | $94.17_{\pm 0.3}$ | $94.70_{\pm 0.43}$ | $95.42_{\pm 0.45}$ |
| Sideburns | $94.80_{\pm 0.3}$ | $95.06_{\pm 0.32}$ | $95.62_{\pm 0.32}$ | $96.33_{\pm 0.47}$ |
| Smiling | $79.45_{\pm 2.54}$ | $81.39_{\pm 1.22}$ | $81.34_{\pm 2.28}$ | $83.56_{\pm 1.77}$ |
| Straight Hair | $80.55_{\pm 0.57}$ | $81.64_{\pm 0.6}$ | $82.95_{\pm 0.75}$ | $84.46_{\pm 0.77}$ |
| Wavy Hair | $76.42_{\pm 0.76}$ | $78.09_{\pm 0.98}$ | $80.24_{\pm 1.35}$ | $82.22_{\pm 0.92}$ |
| Wearing Earrings | $83.72_{\pm 0.59}$ | $84.65_{\pm 0.77}$ | $85.74_{\pm 0.8}$ | $87.08_{\pm 0.75}$ |
| Wearing Hat | $97.07_{\pm 1.46}$ | $97.03_{\pm 1.28}$ | $98.83_{\pm 0.4}$ | $98.95_{\pm 0.25}$ |
| Wearing Lipstick | $90.96_{\pm 0.38}$ | $91.52_{\pm 0.58}$ | $91.87_{\pm 0.58}$ | $92.57_{\pm 0.71}$ |
| Wearing Necklace | $87.93_{\pm 0.41}$ | $88.51_{\pm 0.47}$ | $89.23_{\pm 0.7}$ | $90.37_{\pm 0.59}$ |
| Wearing Necktie | $95.14_{\pm 0.24}$ | $95.56_{\pm 0.33}$ | $96.10_{\pm 0.3}$ | $96.44_{\pm 0.47}$ |
| Young | $83.45_{\pm 1.34}$ | $84.93_{\pm 1.12}$ | $85.79_{\pm 0.78}$ | $87.85_{\pm 0.65}$ |

## B.7 OUT-OF-DISTRIBUTION GENERALIZATION: THE OFFICIAL CELEBA SPLIT

In the main paper, we consider a new CelebA split where train and test sets have overlapping identities. This allows us to test in-distribution performances. In this Subsection, we look at out-of-distribution generalization considering the official CelebA split, where train, validation, and test sets include distinct identities. In other words, we now consider a harder form of generalization by testing agents on never-seen identities at training. Results are shown in Table 8. We observe a similar pattern to the in-distribution results shown in Table 2. That is, we do not observe a benefit for the population, where "best 1 pair" achieves better unseen-generalization than "10 pairs" and "50 pairs". However, when considering imitation and voting, agents develop languages that generalize better to

never-seen identities. The setting "50 pairs + imitation + vote" attains the best performance, with a 93.75% success when communicating about previously unseen identities at training time.

Table 8: Generalization performance on the official CelebA split, in %. In this case, we look at out-of-distribution generalization as train and test sets contain different identities. For each setting we report the mean over 10 seeds. $\pm$ denotes 1 standard error of the mean.

| Setting | best 1 pair | 1 pair | 10 pairs | | | 50 pairs | | |
|---|---|---|---|---|---|---|---|---|
| | - | - | - | +imitation | + imitation+vote | - | +imitation | + imitation+vote |
| Generalization | 92.60 | $90.08_{\pm0.54}$ | $90.42_{\pm0.55}$ | $92.01_{\pm0.40}$ | $92.37_{\pm0.40}$ | $90.91_{\pm0.37}$ | $93.41_{\pm0.19}$ | $\mathbf{93.75}_{\pm0.18}$ |

