# OpenReview forum: "Emergent Communication at Scale"
_ICLR.cc/2022/Conference — ICLR 2022 Spotlight_

### Official Review · Reviewer_HRb2 · 2021-11-02

**Correctness:** 4
**Technical Novelty And Significance:** 3
**Empirical Novelty And Significance:** 4
**Recommendation:** 8
**Confidence:** 3

**Main Review:**

The paper is very well-written. The overall motivation of the work is very clear, and the motivation of each experiments are supported by observations from previous work and theories in cognitive science. The findings are novel and interesting, bringing out the need for conducting large-scale experiments. A minor drawback is that the work largely presents quantitative results and lacks qualitative analyses. At a large scale, are there anything particular about the learned communication that makes it generalize better?

(I didn't aim to write such a short review, but this is a standard exploratory paper that scales up a well-established setting. The experiment setups are well-designed and I don't find any ambiguous point that needs clarification. I am personally not strongly interested in this fully RL communication-learning direction, but researchers that have good reasons for pursuing this topic may find the results very interesting.)

**Summary Of The Paper:**

**After rebuttal**: I am keeping my score

**Before rebuttal**: The paper conducts a large-scale study on RL-based emergent communication. The scale is extended in three dimensions: task complexity, data naturalness, and population size. Scaling up gives new findings that were not observed in previous (small-scale) work. Important findings are (1) evaluating with a large number of candidates is essential for observing the advantage of training with a large number of candidates (2) when experimenting with real images, TopSim, a widely used metric for measuring compositional, no longer correlates with generalization (3) larger population does not necessarily enhance generalization; more clever leverage of the population is needed.

**Summary Of The Review:**

Overall, I find this work thorough and well-motivated. The results would have significant impact in this area. I recommend acceptance. However, I do not work extensively this area and may not follow the latest work.

---

### Official Review · Reviewer_MhrQ · 2021-11-02

**Correctness:** 4
**Technical Novelty And Significance:** 3
**Empirical Novelty And Significance:** 4
**Recommendation:** 8
**Confidence:** 4

**Main Review:**

Strengths:
- This is a timely paper for a problem worth solving.
- I think measuring how well an emergent language transfers to distinct tasks is a very useful metric to determine the utility of an emergent language.
- I appreciate the "best 1 pair" baseline against the population results, it is a very sensible baseline to compare population-based methods against.
- Leveraging population for language emergence is difficult and the presented methods help scale up here while providing some benefits in generalization and robustness.

Weaknesses:
- Although ETL is a useful metric to understand if a language can transfer to a new task, it is useful to measure and understand the structure in the emergent language itself. As indicated in the paper, topographic similarity is probably not the best metric; what about something like TRE [1]? In simple Lewis games, a high topo sim gives a strong clue about what the structure of the language is, but in this case, I have no idea.
- The authors construct test sets that have overlapping identities for CelebA. While this makes sense for the ETL experiment they set up, in general, this is ensuring that the train and test splits are more in-distribution, making the evaluation task much easier. One of the most important questions w.r.t. emergent languages is if they can generalize systematically. To this end, I would prefer a systematic split where one can study how the emergent language performs in a systematically out-of-distribution setup. An example of something like this would be including CelebA samples that contain a larger proportion of certain types of correlations (e.g. bald with beard) in the train set but have uncorrelated attributes for distinct identities in the test set.

Questions:
- You observe that scaling up the task difficulty entails unstable RL optimization. Did you try other ways to backpropagate the gradients to the speaker such as Gumbel straight-through estimation?
- What is the significance of "ease" in "Ease and transfer learning"? Do you measure ease of learning, e.g. through acquisition speed or something else?

Suggestions:
- Section 1: Typo in the second to last line of Introduction: "emergent" -> "emergence".
- I would suggest adding a period (.) at the end of paragraph titles (between the title and the paragraph content).

References:
[1] Andreas, J., 2019. Measuring compositionality in representation learning. arXiv preprint arXiv:1902.07181.

**Summary Of The Paper:**

This paper aims to scale up the usual Lewis game language emergence setup. Instead of simple synthetic objects, the authors consider scaling up to image datasets such as ImageNet and CelebA. Another aspect they scale up is the task difficulty, by increasing the number of candidates the listener has to choose the correct answer from. Third, they also attempt scaling up the population size. To deal with the instability arising from a large number of candidates, the authors use established stabilization techniques such as minimization of KL divergence between the online policy and an exponentially moving average target policy. The paper also introduces a new metric called Ease and transfer learning (ETL) to measure how well an emergent language transfers to a novel task as an alternative to topographic similarity. Finally, the authors explore some techniques such as imitation learning and inference-time voting to try to leverage a population of agents.

**Summary Of The Review:**

The commonly used Lewis game setups with synthetic data for language emergence are indeed quite limited for real-world applicability. Thus, scaling up the game for the emergence of communication protocols is an important problem. I think this paper is a great contribution, but I would appreciate some more understanding of the structure within the emergent languages. Alternatively (or additionally), demonstrating an ability to systematically generalize can also be very valuable.

---

> ### Author Response · Authors · 2021-11-12
> **Response to Reviewer MhrQ**
>
> > *One of the most important questions w.r.t. emergent languages is if they can generalize systematically. To this end, I would prefer a systematic split where one can study how the emergent language performs in a systematically out-of-distribution setup.*
>
> Thank you for the suggestion. We add in Appendix B.7 of the updated version the generalization performances for different population sizes when considering the official CelebA split. The latter, as you suggested, include non-overlapping identities between train, validation, and test sets. In other words, we now consider a harder form of generalization by testing agents on never-seen identities at training. We observe in the Appendix a similar pattern to the in-distribution results. That is, we do not note a benefit for the population, where ''best 1 pair'' achieves better unseen-generalization than ''10 pairs'' and ''50 pairs''. However, when considering imitation and voting, agents develop languages that generalize better to never-seen identities. In summary, we did not observe a significant difference between the official zero-shot split and our proposed split.
>
> > *You observe that scaling up the task difficulty entails unstable RL optimization. Did you try other ways to backpropagate the gradients to the speaker such as Gumbel straight-through estimation?*
>
> As we scale up emergent communication experiments, the Gumbel straight-through estimator may indeed be more robust than RL optimization, and it has been used in plenty of previous research in the field (Jang et al, 2017, Maddison et al, 2017, Mordatch and Abbeel, 2018, Lowe et al, 2020). That said, we focus here on the RL approach as it is more generic and arguably more aligned with the constraints of human communication wherein there is no exact error backpropagation between speakers and listeners. Furthermore, our preliminary results with simple tasks showed that RL optimization works better for a sequence of symbols.
>
> > *What is the significance of "ease" in "Ease and transfer learning"? Do you measure ease of learning, e.g. through acquisition speed or something else?*
>
> Exactly! By ease, we refer to the acquisition speed of a given task.
>
>
> > *Section 1: Typo in the second to last line of Introduction: "emergent" -> "emergence".
> I would suggest adding a period (.) at the end of paragraph titles (between the title and the paragraph content).*
>
> Thank you. We updated the paper accordingly.

---

> > ### Comment · Reviewer_MhrQ · 2021-11-13
> > **Thank you**
> >
> > Thank you for your response.
> >
> > I really appreciate the experiments on CelebA without overlapping identities. However, I imagine the models are more likely picking up on various facial attributes in the faces than the actual identities. For a true systematic split, one would need to hold out certain combinations of attributes to see if the model can generalize to them during evaluation. This is perhaps not straightforward to set up, and the lack of such a study does not affect my score for this paper.

---

### Official Review · Reviewer_zVAw · 2021-11-03

**Correctness:** 3
**Technical Novelty And Significance:** 3
**Empirical Novelty And Significance:** 4
**Recommendation:** 8
**Confidence:** 4

**Main Review:**

**Pros**

1. Thorough and balanced discussion of benefits/drawbacks among the scaling dimensions.
2. Research is very well-situated in the literature.
3. Experiments are well-motivated, novel to my knowledge, and support the authors' case to scale up emergent communication experiments.
4. Reads very clearly, especially the model and experiment implementation. The technical implementation details in Section 2 are precise.

**Cons**

1. Questions about statistical rigor, especially concerning few datapoints (1, 3 in additional comments)

**Questions and Additional Comments**

(1) The main paper considers |C| = 16, 64, 256, 1024 (the appendix considers also |C| = 4096) and N = 1, 10, 50. It is hard to justify conclusions such as "increased task complexity systematically improves generalization and ETL" and "there is no consistent pattern between TopSim and |C| with a non-significant Spearman correlation" with 4 datapoints. Were intermediate datapoints tested but not reported?

(2) In Sections 3.1 and 3.2 (Fig. 2, Table 1), several metrics such as accuracy, generalization, and ETL are shown as functions of |C| (task complexity). However, |C| grows exponentially while the performance improvement in these metrics grows roughly linearly. In Table 2, going from 10 to 50 pairs requires roughly 5x more training hours (max is 20 days), whereas their mean performance may not be significantly different. Do the authors think the cost of scaling up is worth the diminishing returns?

(3) In Section 3.2 (TopSim may fall short with natural images): "As the latter correlates... we expect to find a correlation between TopSim and |C| by transitivity." The authors imply that |C| ~ generalization and TopSim !~ |C| (Fig 3) ---> TopSim !~ generalization. In general, transitivity of correlation is not guaranteed. (For example, take X and Y are independent and X + Y = Z.) I would avoid this argument and replace it by directly computing the correlation between TopSim and generalization accuracy.

In addition, "...TopSim and |C| with a ... between TopSim and generalization for ImageNet and CelebA respectively". I found this sentence hard to parse-- are the reported correlations between TopSim and |C| or TopSim and generalization?

(4) It would improve the focus of the paper to state the work's relationship to the cognitive science and the RL sides of emergent communication more explicitly. After reading the paper, it is clear that the authors focus on the robotics/modelling aspect of emergent communication rather than the evolutionary origins of human language. However, the authors begin by discussing both [cognitive science/evolutionary linguistics] and RL as reasons to scale up. It's not clear that starting with complex tasks is the best model of human language emergence. Work in cognitive science theorizes that from both an evolutionary and a first language acquisition perspective, complex language develops from simple communication such as pointing and pantomiming in simple settings rather than from immediately solving difficult tasks (see Barrett and Skyrms 2017, Deacon 1998, Stern 1974). Given the authors partially motivated their work with researching the origins of human language, it would be nice to see more discussion on how their experiments mirror (or contradict) hypothesized settings of language development. For example, in Section 3.3's voting, this is a technique that improves model performance but is likely not how language originated in humans. Alternatively, this can be avoided if the authors state that their work just focuses on the RL side.

**Summary Of The Paper:**

The paper argues for scaling up experiments in emergent communication to better understand the evolution of human language and to construct more efficient representations. To achieve this, the authors demonstrate scaling up along dataset, task complexity, and population size in the context of a multi-agent Lewis game, evaluating the benefits and drawbacks of each using generalization, robustness to input noise, and ETL. They find that (1) KL regularization mitigates unstable optimization when scaling up; (2) increasing task complexity improves generalization and better separates performance of different algorithms; (3) topsim and generalization are uncorrelated, necessitating alternative metrics of compositionality at scale; (4) while scaling up population in itself is not obviously beneficial, imitation learning and voting in larger populations produce more "robust, productive, and easy-to-learn" languages. Overall, the paper makes a strong case to scale up emergent communication experiments supported by empirical contributions.

**Summary Of The Review:**

**Recommendation**

I recommend to accept this paper based on its valuable experimental results, which demonstrate that scaling up emergent communication can find more efficient and generalizable representations. Supported by empirical contributions, the paper steps beyond toy experiments that are common in emergent communication literature and towards increasingly complex and realistic settings.

---

> ### Author Response · Authors · 2021-11-12
> **Response to Reviewer zVAw**
>
> > *It is hard to justify conclusions [...] with 4 datapoints. Were intermediate datapoints tested but not reported?*
>
> Actually we tested more datapoints, varying |C| in {16, 32, 64, 128, 256, 512,1024} when looking at TopSim and generalization correlation, with 10 seeds for each setting. We still do not note a significant correlation when considering these intermediate points.
>
> > *|C| grows exponentially while the performance improvement [...] grows roughly linearly. [...] going from 10 to 50 pairs requires roughly 5x more training hours, whereas their mean performance may not be significantly different. Do the authors think the cost of scaling up is worth the diminishing returns?*
>
> Thank you for this question! The short answer is that some scaling up, e.g., |C| can often be done without significant computational costs; that population scaling is indeed expensive; and that the optimal trade-off depends on the research goal. First, scaling up the task complexity exponentially leads to a roughly linear improvement in our metrics. This evolution should even saturate at some point; since we measure bounded metrics against an unbounded parameter, we will necessarily observe diminishing returns (we cannot get to a precision higher than 100%, or a reconstruction loss lower than 0). Note though that, for a fixed and reasonable batch size value, increasing |C| does not necessarily lead to more costly computation. Indeed, the set of candidates C is chosen randomly from the batch, X. In our implementation, the hardest possible task, when |C|=|X|, is constructed by considering the entire batch as candidates, making the training even faster since we avoid the sampling step. For example, in Fig2.(b), experiments with |C|=1024 are the fastest to train while leading to languages with the highest generalization performances at test time. However, we do agree that considering very large |C| values (and hence |X|) would require significant computation. In this case, we should instead think about other ways to complexify the task such as considering harder negatives (see also discussion with Reviewer YDs6).
> Second, larger population size increases considerably the computational cost (see Appendix B.2). While more efficient implementation may mitigate this constraint, balancing performance and computation cost is a relevant question. To partially address it, we examine the “best 1 pair” setting that asks if the cost of scaling up from 1 to 10 pairs is worth the diminishing returns. However, as you correctly mentioned, it ignores settings with larger N, and does not fully address the question of 1 vs. 10 pairs.  We believe that this trade-off in the population setting case between computational cost and more efficient communication protocol depends on the research goal.
>
> > *In general, transitivity of correlation is not guaranteed. I would avoid this argument and replace it by directly computing the correlation between TopSim and generalization accuracy.*
>
> Thank you for the comment. We updated the text accordingly. We now directly report the correlation between TopSim and generalization accuracy.
>
> > *Are the reported correlations between TopSim and |C| or TopSim and generalization?*
>
> We report the correlation between TopSim and generalization. We clarified this comment in the updated version of the paper in the paragraph “TopSim may fall short with natural images”.

---

> > ### Author Response · Authors · 2021-11-19
> > **Response to Reviewer zVAw Part 2**
> >
> > > It would improve the focus of the paper to state the work's relationship to the cognitive science and the RL sides of emergent communication more explicitly. After reading the paper, it is clear that the authors focus on the robotics/modelling aspect of emergent communication rather than the evolutionary origins of human language. However, the authors begin by discussing both [cognitive science/evolutionary linguistics] and RL as reasons to scale up. It's not clear that starting with complex tasks is the best model of human language emergence. Work in cognitive science theorizes that from both an evolutionary and a first language acquisition perspective, complex language develops from simple communication such as pointing and pantomiming in simple settings rather than from immediately solving difficult tasks (see Barrett and Skyrms 2017, Deacon 1998, Stern 1974). Given the authors partially motivated their work with researching the origins of human language, it would be nice to see more discussion on how their experiments mirror (or contradict) hypothesized settings of language development. For example, in Section 3.3's voting, this is a technique that improves model performance but is likely not how language originated in humans. Alternatively, this can be avoided if the authors state that their work just focuses on the RL side.
> >
> >
> > Thank you for your comments. There are indeed different theories about the necessity of complex tasks to model human communication (e.g., Barret and Skyrms 2017 vs. Bickerton 2015, Dupoux 2018). We endorse here the second view and rely on performance-inspired solutions in order to scale up.
> > We updated our conclusion to clarify this confusion. In particular, we added these sentences:
> >
> > *“There are different theories about the necessity of complex tasks to model human communication (e.g., Barret and Skyrms 2017 vs. Bickerton 2015, Dupoux 2018). We endorse here Bickerton’s view and adopt performance-inspired solutions to scale up, such as KL-regularization and imitation. An interesting future debate is whether our findings here could influence the progress of similar research in human communication.”*

---

> > > ### Comment · Reviewer_zVAw · 2021-11-29
> > > **Response**
> > >
> > > Thank you for your comments. I will keep my score of 8 for the paper.

---

### Official Review · Reviewer_YDs6 · 2021-11-06

**Correctness:** 4
**Technical Novelty And Significance:** 3
**Empirical Novelty And Significance:** 4
**Recommendation:** 8
**Confidence:** 3

**Main Review:**

Strengths:
 - Emergent communication has classically focused on toy tasks, so a scaling paper is a much-needed contribution
 - Overall the paper is quite clear and well-written, with clear notation
 - Analysis of KL regularization in the emergent communication domain is interesting and important
 - Analysis of harder training tasks is also compelling; I wonder if you also add "hard negatives" to make the task more difficult
 - ETL is a useful proposed metric, albeit partially based on existing work
 - Best single-pair seed is indeed the correct metric for population-based work, which is missed in prior work (Cogswell, et al.)
 - Analysis of population size is interesting and a compelling rebuttal to previous work

Weaknesses:
 - Not clear to me that a lack of correlation between TopSim and |C| implies that TopSim is not a useful metric for generalization; although |C| does correlate with generalization, I think the usual argument is that TopSim correlates with compositional generalization to specific types of out-of-domain examples (i.e., those which have combinations of previously seen attributes of training examples). The finding is still interesting, but it might be worth tempering the claim here a bit.
 - Given that the paper argues for scaling up a field, more information about compute usage and hardware would be appreciated

**Summary Of The Paper:**

This paper provides an analysis of emergent communication behavior at scale and discusses training techniques to reduce instability during RL optimization.

**Summary Of The Review:**

I strongly recommend accepting this paper, given the over-emphasis of toy tasks in the emergent communication community.

---

> ### Author Response · Authors · 2021-11-12
> **Response to Reviewer YDs6**
>
> >*I wonder if you also add "hard negatives" to make the task more difficult*
>
> This is indeed a very interesting direction to explore in future works. Studies in emergent communication (e.g., [1]) and representation learning (e.g., [2, 3])  suggest that “how” negatives are chosen impacts the learned representation. For example, [1] shows that selecting more realistic negatives biases the communicative protocol making it richer, while [3] demonstrates that hard negatives lead to “faster and better learning”. We believe that such observation would hold in our framework. However, systematic experiments should be conducted in future works.
>
>
> >*Given that the paper argues for scaling up a field, more information about compute usage and hardware would be appreciated*
>
>  Thank you for your comment. Note that we report in Appendix B.2(Table 5) some computational requirements for our base setup. Specifically, we report the memory usage, time per step, and the total training time when considering common hardware (p100 and v100). We’ve slightly reworded Table 5 to make it clearer that the “Memory” column refers to GPU memory usage, and not RAM. Since our pipeline uses very light preprocessing, host CPU/RAM usage is mostly irrelevant. If you have other useful usage metrics in mind, we would be happy to include them.
>
> [1] Lazaridou, A., Hermann, K. M., Tuyls, K., & Clark, S. (2018). Emergence of linguistic communication from referential games with symbolic and pixel input. arXiv preprint arXiv:1804.03984.
>
> [2] Robinson, J., Chuang, C. Y., Sra, S., & Jegelka, S. (2020). Contrastive learning with hard negative samples. arXiv preprint arXiv:2010.04592.
>
> [3] Kalantidis, Y., Sariyildiz, M. B., Pion, N., Weinzaepfel, P., & Larlus, D. (2020). Hard negative mixing for contrastive learning. arXiv preprint arXiv:2010.01028.

---

> > ### Author Response · Authors · 2021-11-19
> > **Response to Reviewer YDs6 Part 2**
> >
> > > Not clear to me that a lack of correlation between TopSim and |C| implies that TopSim is not a useful metric for generalization; although |C| does correlate with generalization, I think the usual argument is that TopSim correlates with compositional generalization to specific types of out-of-domain examples (i.e., those which have combinations of previously seen attributes of training examples).
> >
> > Thank you for your comment. In the updated version of the paper, we clarify this statement by directly reporting the correlation between TopSim and the generalization accuracy (see also the response for Reviewer zVAw). Our results show no significant correlation between TopSim and generalization (Sec 3.2, p>0.19 and p=0.13 for CelebA pretrained logits and attribute representations, respectively). In our case, generalization is computed by reporting agents' accuracy when communicating about never-seen in-domain examples at test time. These examples, even if not out-of-domain, are new combinations of previously seen attributes at training time. For instance, in the simple case of disentangled attributes, agents could learn to communicate about red circles and blue triangles at training, then be tested on blue circles. This type of generalization is believed to benefit from high TopSim (see e.g., [1,2,3]). We clarify this in our updated version of the paper (see the discussion on language properties in Section 2.4).
> >
> > [1] Lazaridou, A., Hermann, K. M., Tuyls, K., & Clark, S. (2018). Emergence of linguistic communication from referential games with symbolic and pixel input. arXiv preprint arXiv:1804.03984.
> >
> > [2] Kirby, S., Cornish, H., and Smith, K. (2008). Cumulative cultural evolution in the laboratory: An experimental approach to the origins of structure in human language. Proceedings of the National Academy of Sciences
> >
> > [3] Li, F., & Bowling, M. (2019). Ease-of-teaching and language structure from emergent communication. arXiv preprint arXiv:1906.02403.

---

### Author Response · Authors · 2021-11-12
**General response**

We would like to thank all the reviewers for their time and constructive feedback! Please find answers to some of your comments/questions below.

---

### Author Response · Authors · 2021-11-19
**Qualitative analysis/structure understanding of the messages: response to reviewers MhrQ  and HRb2**

Thank you for all your constructive feedback. Both reviewers MhrQ and HRb2 mentioned that qualitative analysis could provide a better understanding of the emergent languages’ structure. We also believe that understanding the structure of the emergent language is very important. However, as you may know, qualitative evaluation can be a complicated question in emergent communication. In this context, one of the reasons for small-scale experiments (small vocabulary, short message length) is to be able to humanely analyze the developed languages, e.g., Fig4 [1], Appendix B [2]). As soon as we move to complex domains with non-trivial communication channels, qualitative analysis is more challenging. In our case, we considered face reconstruction as a qualitative analysis (see Fig.3(right) and Appendix B.1). The latter shows that more complex tasks lead to more accurate reconstruction, encoding more fine-grained information such as the presence of head cover or eye glasses. Interestingly, a concurrent ICLR paper explored natural language translation with emergent protocol as a qualitative analysis of emergent languages [3]. We also explored other measures to grasp the languages' structure such as the within-distance when clustering messages per identities in CelebA. Unlike the face reconstruction experiment, the clustering measure showed no difference between our settings. This suggests that, when faced with complex tasks, agents do not communicate about identities but rather become better at discriminating between different attributes (see Appendix B.1). As these attributes may or may not be encoded in CelebA, a non-visual analysis, unlike ours, would not be informative. If you have further potential ideas for qualitative analyses of languages, we would be happy to add them to the next version of the paper.


[1] Cogswell, Michael, et al. "Emergence of compositional language with deep generational transmission." arXiv preprint arXiv:1904.09067 (2019).

[2] Ren, Yi, et al. "Compositional languages emerge in a neural iterated learning model." arXiv preprint arXiv:2002.01365 (2020).

[3] Anonymous Linking Emergent and Natural Languages via Corpus Transfer. Under review ICLR.

---

### Decision · Program_Chairs · 2022-01-20

**Decision:**

Accept (Spotlight)

**Comment:**

This manuscript expands the range of recent work in reinforcement learning for language games to much larger and more realistic datasets. A timely and relevant contribution and one that is well evaluated. Further work in stabilizing RL approaches for such large-scale problems is likely to have other far-reaching consequences. Reviewers were unanimous that this is a strong submission after the author discussion period.